# Shortcut-connected Expert Parallelism for Accelerating Mixture of Experts

**Weilin Cai** [1]   **Juyong Jiang** [1]   **Le Qin** [1]   **Junwei Cui** [1]   **Sunghun Kim** [1]   **Jiayi Huang** [1]

## Abstract

Expert parallelism has emerged as a key strategy for distributing the computational workload of sparsely-gated mixture-of-experts (MoE) models across multiple devices, enabling the processing of increasingly large-scale models. However, the *All-to-All communication* inherent to expert parallelism poses a significant bottleneck, limiting the efficiency of MoE models. Although existing optimization methods partially mitigate this issue, they remain constrained by the sequential dependency between communication and computation operations. To address this challenge, we propose ScMoE, a novel shortcut-connected MoE architecture integrated with an overlapping parallelization strategy. ScMoE decouples communication from its conventional sequential ordering, enabling up to 100% overlap with computation. Compared to the prevalent top-2 MoE baseline, ScMoE achieves speedups of $1.49\times$ in training and $1.82\times$ in inference. Moreover, our experiments and analyses indicate that ScMoE not only achieves comparable but in some instances surpasses the model quality of existing approaches.

## 1. Introduction

In recent years, Transformer-based large language models (LLMs) have significantly propelled the fields of Natural Language Processing (Vaswani et al., 2017; Brown et al., 2020; Wei et al., 2022b; Ouyang et al., 2022; Wei et al., 2022a; Chowdhery et al., 2023; Achiam et al., 2023), Computer Vision (Dosovitskiy et al., 2021; Liu et al., 2021), and Multimodality (Lu et al., 2019; Zhou et al., 2022; Zhang et al., 2021; Zhu et al., 2023). The sparsely-gated mixture-of-experts (MoE) approach has been integral in increasing parameter counts and enhancing model performance across various modalities (Shazeer et al., 2017; Riquelme

et al., 2021; Mustafa et al., 2022; Jiang et al., 2024). Expert parallelism (Lepikhin et al., 2021; Fedus et al., 2022) has emerged as a viable strategy to efficiently distribute MoE computations over multiple devices, synergizing with conventional parallelism techniques (Hwang et al., 2023; Singh et al., 2023) such as data parallelism (Rajbhandari et al., 2020; 2021) and model parallelism (Narayanan et al., 2021; Smith et al., 2022).

Nevertheless, expert parallelism incurs substantial *All-to-All communication* overhead (Lepikhin et al., 2021; Fedus et al., 2022), which can contribute to approximately 50% of the total time in intra-node multi-GPUs or multi-nodes distributed environments (see Figure 1), thus forming a critical bottleneck in scaling MoE models (Nie et al., 2022; Hwang et al., 2023; Mayer & Jacobsen, 2020; Smith et al., 2022). Despite existing optimizations such as hierarchical All-to-All (He et al., 2022; Nie et al., 2022) and pipelining (Hwang et al., 2023; Zhang et al., 2023) strategies that mitigate communication delays and partially overlap communication with computation, the communication challenge persists due to the inherent sequential dependencies between these operations (Wang et al., 2023). To address this constraint, our intuitive idea is to reconstruct the inputs of MoE layer by incorporating not only the current-layer but also the preceding-layer representations through a shortcut connection, thereby refining the communication-computation dependencies and expanding the potential for their overlapping optimization.

In this paper, we propose the shortcut-connected MoE (ScMoE) architecture, which completely decouples communication processes from the sequence of conventional MoE models. ScMoE architecture is initially built on the standard top-2 MoE (see Figure 2 (a)), which typically substitutes the multi-layer perceptron (MLP) module with a top-2 gating MoE module in every second Transformer block (refer to the Transformer block with the MoE module as "current layer", and the preceding one without the MoE module as "preceding layer"). Diverging from the top-2 approach, ScMoE utilizes a top-1 MoE module to process preceding-layer representations via a shortcut connection, while employing a shared expert (an MLP module) to process current-layer representations. These two processes are independently managed in parallel, with their results integrated into the final output of the current layer. Furthermore, the ScMoE

[1]The Hong Kong University of Science and Technology (Guangzhou). Correspondence to: Jiayi Huang <hjy@hkust-gz.edu.cn>.

*Proceedings of the $42^{nd}$ International Conference on Machine Learning*, Vancouver, Canada. PMLR 267, 2025. Copyright 2025 by the author(s).

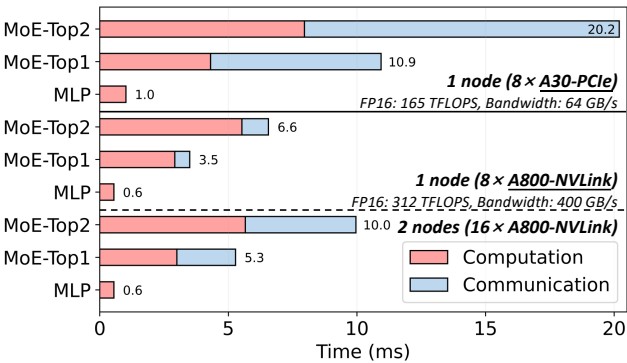

*Figure 1.* The overhead of MLP and top-2/top-1 MoE in a transformer block of SwinV2-MoE-S (Hwang et al., 2023) model, allocating one expert per GPU with expert parallelism. The All-to-All communication takes up 60% of total time on a single node with 8×A30 GPUs, but drops to 15% on 8×A800 due to the latter's 6× higher bandwidth provided by GPU-to-GPU NVLink (Foley & Danskin, 2017). Despite benefiting from NVLink, communication still approaches 50% due to the lower-bandwidth inter-node Ethernet (Li et al., 2020) when scaling across multiple nodes.

architecture can be extended to accommodate MoE models that employ various MoE placement frequencies, such as integrating an MoE module into every Transformer block.

To efficiently overlap the decoupled communication and computation within the ScMoE architecture, we implement an adaptive overlapping parallel strategy that dynamically schedules operators based on actual performance metrics. Compared to existing optimization strategies such as pipelining (Huang et al., 2019; Hwang et al., 2023), our approach not only doubles the overlap duration, but also realizes complete overlapping of communication in scenarios where communication time does not exceed the computation duration. Furthermore, our method essentially advances the MoE architecture in algorithm aspect, which is device-agnostic to improve the efficiency of MoE model, thus ensuring a broad applicability across various hardware configurations and maintaining compatibility with current optimizations.

The experimental results reveal that, compared to the standard top-2 MoE, our proposed ScMoE architecture optimally accelerates training by $1.49\times$ and $1.14\times$ in 8×A30-PCIe and 8×A800-NVLink scenarios characterized by high and low communication overheads, respectively, and accelerates inference by $1.82\times$ and $1.21\times$. Moreover, we perform experiments on different configurations of the ScMoE architecture, including shortcut-connected position and coefficient gating network. Considering the optimal accuracy and the relatively longer overlap duration, we favor selecting the intermediate representations between the Attention and MLP modules in the preceding layer as the input for the gate-routed experts. In addition, ScMoE has been demonstrated through experiments and theoretical analysis to attain or exceed the model quality of existing methods

in both vision and language downstream tasks. We also conduct an in-depth analysis and discussion of the ScMoE architecture, investigating the proposed shortcut connection and exploring opportunities for further development.

In summary, our contributions are as follows:

- We propose the shortcut-connected MoE (ScMoE) architecture that breaks the conventional dependency between communication and computation in distributed MoE models, bypassing the restrictions imposed on current communication optimization techniques.

- We develop an adaptive overlapping strategy for advancing expert parallelism with the shortcut-connected MoE, which significantly improves the efficiency of MoE models and ensures broad compatibility.

- We conduct empirical evaluation and theoretical analysis on our methods, confirming that our methods accelerate MoE models while achieving comparable or even better model quality compared to existing methods, and offer in-depth analysis and discussion on the effectiveness of the proposed shortcut connection.

## 2. Background & Related Work

### 2.1. Sparsely-Gated Mixture of Experts

The sparsely-gated mixture-of-experts (Shazeer et al., 2017) (MoE) layer is composed of multiple multi-layer perceptron (MLP) sub-networks, termed "experts," and employs a trainable gating network to selectively activate a subset of these experts during each iteration. Given $N$ expert networks $\{E_i\}_1^N$, gating network $G$ and input representation $x$, the output of MoE module can be written:

$$MoE(x) = \sum_{i=1}^{N} G(x)_i E_i(x). \tag{1}$$

Following the prevailing approach in existing MoE research, we use the noisy top-k softmax gating network to select $k$ experts for the computation, formalized by

$$G(x) = Softmax(\overline{TopK}(H(x), k)), \tag{2}$$

$$\overline{TopK}(H(x), k)_i = \begin{cases} H(x)_i, & \text{if } H(x)_i \in TopK(H(x)). \\ -\infty, & \text{otherwise.} \end{cases} \tag{3}$$

$$H(x)_i = (x \cdot W_{gate})_i + \epsilon_i, \tag{4}$$

$$\epsilon_i = StandardNormal() \cdot Softplus((x \cdot W_{noise})_i), \tag{5}$$

where $\epsilon$ is tunable Gaussian noise, $W_{gate}$ and $W_{noise}$ denote two trainable weight matrices.

Leveraging sparse output of $G(x)$, this approach significantly increases the number of model parameters without

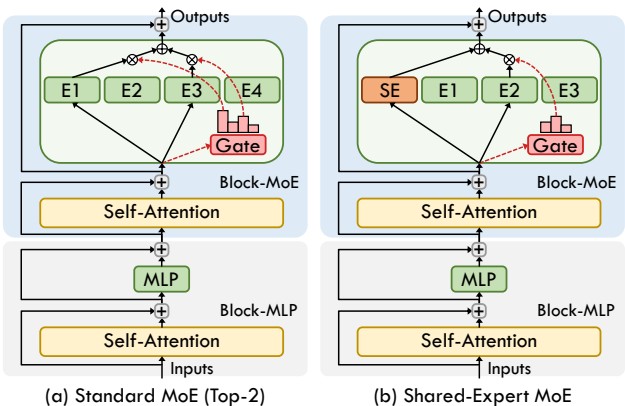

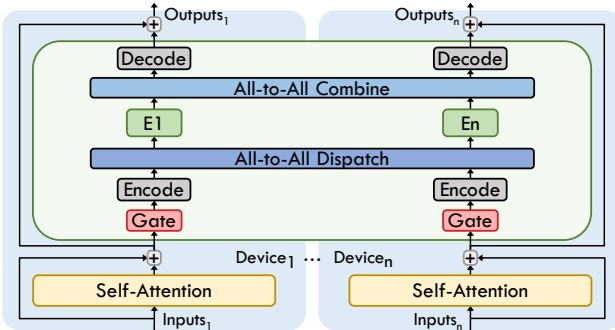

Figure 3. Illustration of scaling MoE transformer layer across multiple devices with expert parallelism.

*Figure 2.* Illustrations of the standard top-2 MoE architecture (a) and the corresponding shared-expert MoE architecture (b). "SE" in (b) denotes the shared expert.

causing a proportional increase in computational demand. The value of $k$ can be set to 1 or 2 or even higher values. Opting for a larger $k$ moves the model closer to the dense architecture, which generally results in higher prediction accuracy (Riquelme et al., 2021), but also leads to greater computational overhead.

Figure 2 (a) illustrates the prevailing top-2 MoE architecture. Each Transformer block with MoE module, denoted by a light blue block and referred to as "Block-MoE," replaces the MLP with a set of experts ("E1, E2, E3, E4") and a gating network ("Gate"). Following prior work (Lepikhin et al., 2021; Du et al., 2022; Shen et al., 2023; Hwang et al., 2023; Lieber et al., 2025), the "Block-MoE" is interspersed with the conventional Transformer block, depicted as a gray block and referred to as "Block-MLP." Additionally, various options exist for the frequency of MoE module placement, including placing an MoE module into every block (Jiang et al., 2024; Dai et al., 2024; Qwen, 2024; Databricks, 2024) or every four blocks (Zoph et al., 2022; Xue et al., 2024).

**Shared Expert.** In contrast to the standard top-2 MoE architecture, the shared-expert MoE incorporates a fixed dense MLP module to process all input tokens, combining its output with the result from the top-1 gating expert for each token, as illustrated in Figure 2 (b). Given the shared expert $SE$, the output of the MoE module is formulated as:

$$MoE(x) = SE(x) + \sum_{i=1}^{N} G(x)_i E_i(x). \qquad (6)$$

This method, originally proposed by DeepSpeed-MoE (Rajbhandari et al., 2022), activates the same number of experts as the standard MoE for computation while reducing dynamic expert selection and communication volume. Extensive empirical results from DeepSpeed-MoE and subsequent studies (Dai et al., 2024; Qwen, 2024) demonstrate that the shared expert architecture achieves model quality that is on

par with or even superior to existing approaches, leading to its growing adoption (Xue et al., 2024; Wu et al., 2023; Chen et al., 2024; Gou et al., 2023; Gao et al., 2024).

### 2.2. Expert Parallelism

To facilitate efficient distributed training and inference of MoE models, expert parallelism is proposed to allocate unique experts to each distributed computing device such as GPU and TPU, and map tokens to their corresponding experts through All-to-All communication across participating devices (Lepikhin et al., 2021; He et al., 2021; Nie et al., 2022). As illustrated in Figure 3, the workflow of MoE employing expert parallelism is segmented into the following sequential operations: gate routing, input encode, All-to-All dispatch, expert computation, All-to-All combine, and output decode. To enhance the efficiency, input encode is employed to aggregate the token data layout to a contiguous format before All-to-All dispatch, and output decode is the inverse process after All-to-All combine. Furthermore, the integration of expert parallelism with other parallelisms (Hwang et al., 2023; Singh et al., 2023; Fedus et al., 2022; Zheng et al., 2022) has been explored to support the scaling of larger MoE models on extensive distributed systems. However, the All-to-All communication used for token transfer has been a primary bottleneck limiting the efficiency of distributed MoE models, as shown in Figure 1.

### 3. Shortcut-connected MoE Designs

In the prevailing Transformer-based model, the MoE module substitutes MLP to sequentially manipulate intermediate representations (Lepikhin et al., 2021; Du et al., 2022; Shen et al., 2023), impeding the efficacy of existing optimization strategies (He et al., 2022; Nie et al., 2022; Hwang et al., 2023; Zhang et al., 2023) due to the limited interaction within the MoE module. To address the aforementioned limitations, we propose the shortcut-connected MoE (ScMoE) architecture, which enables optimization opportunities for computation-communication overlap for expert parallelism.

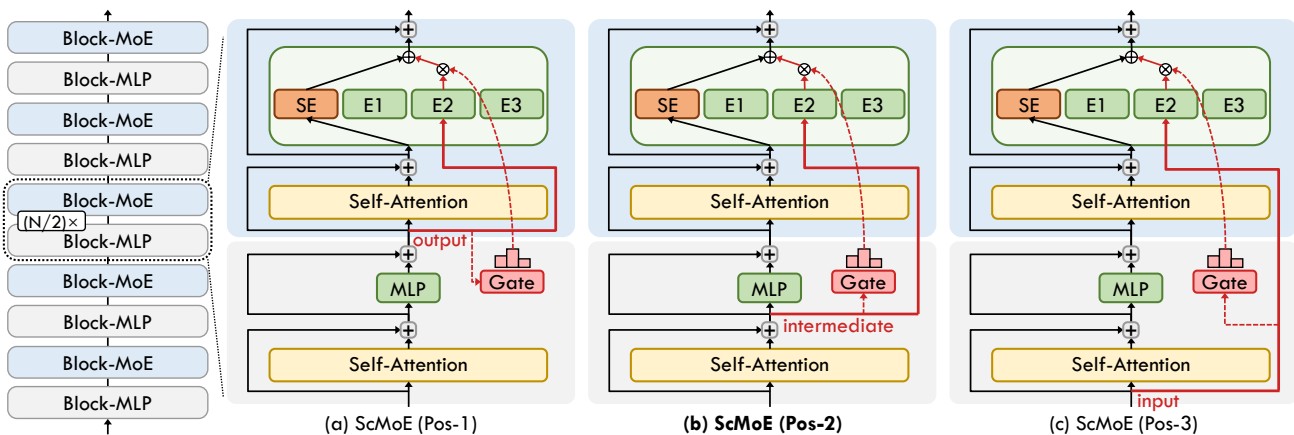

*Figure 4.* Illustrations of various ScMoE architectures with shortcut connections to different positions of the preceding layer: (a) "Pos-1" output, (b) "Pos-2" intermediate, and (c) "Pos-3" input. The red line indicates the transmission of the preceding-layer representations to the MoE via a shortcut connection. Details regarding pre-layer normalization and dropout procedures have been excluded for simplicity.

## 3.1. Architectural Design

In this section, we introduce the shortcut-connected MoE (ScMoE) architecture. Unlike the prevailing MoE architectures, illustrated in Figure 2, which focus on processing intermediate representations within the current layer (the Transformer block containing the MoE), the ScMoE processes representations from both the current and preceding layers. Specifically, ScMoE employs a top-1 MoE module to handle representations from the preceding layer via a shortcut connection, while a shared expert processes the current-layer representations. These two operations are conducted independently, with their outcomes integrated into the final output of the current layer, facilitating communication and computation overlap between these two processes.

While the shared expert processes the same intermediate representations in the current layer as the prevailing MoE approaches, we explore three distinct preceding-layer representations for ScMoE's top-1 MoE process, as illustrated in Figure 4. The configurations "Pos-1" (a), "Pos-2" (b), and "Pos-3" (c) represent shortcuts connecting different positions of the preceding layer: output, intermediate, and input, respectively. Given that $\mathcal{T}_{Atten}$, $\mathcal{T}_{SE}$, and $\mathcal{T}_{MLP}$ represent the durations of Attention, Shared Expert, and MLP, respectively, the corresponding overlap durations are (a) $\mathcal{T}_{Atten} + \mathcal{T}_{SE}$, (b) $\mathcal{T}_{Atten} + \mathcal{T}_{SE} + \mathcal{T}_{MLP}$, (c) $2\mathcal{T}_{Atten} + \mathcal{T}_{SE} + \mathcal{T}_{MLP}$.

Using the "Pos-2" configuration as an example, this ScMoE architecture can be formulated as follows:

*Block-MoE:*

$$\mathcal{H}_{l+1}^{\text{ScMoE}} = \mathcal{H}_{l+1}^{MH} + SE^{(l+1)}(\mathcal{H}_{l+1}^{MH})$$
$$+ \sum_{i=1}^{N} G(\mathcal{H}_l^{MH})_i E_i(\mathcal{H}_l^{MH}), \quad (7)$$

$$\mathcal{H}_{l+1}^{MH} = \mathcal{H}_l^{MLP} + \text{MultiHead}_{MoE}^{(l+1)}(\mathcal{H}_l^{MLP}), \quad (8)$$

*Block-MLP:*

$$\mathcal{H}_l^{MLP} = \mathcal{H}_l^{MH} + \text{MLP}^{(l)}(\mathcal{H}_l^{MH}), \quad (9)$$

$$\mathcal{H}_l^{MH} = \mathcal{H}_{l-1} + \text{MultiHead}_{MLP}^{(l)}(\mathcal{H}_{l-1}), \quad (10)$$

where $\mathcal{H}_{l+1}^{\text{ScMoE}}$ refers to the output from the MoE sub-layer, $\mathcal{H}_{l+1}^{MH}$ signifies the output from the Multi-Head Attention (MultiHead) sub-layer MultiHead$_{MoE}^{(l+1)}(\cdot)$ in the $(l+1)$-th Transformer block ("Block-MoE"). $SE^{(l+1)}(\cdot)$ denotes the shared expert while $E_1, ..., E_N$ represent the $N$ gate-routed experts. The gating network $G(\cdot)$ is referred to as Equation 2. $\mathcal{H}_l^{MLP}$ and $\mathcal{H}_l^{MH}$ are the outputs of the MLP sub-layer MLP$^{(l)}(\cdot)$ and the MultiHead sub-layer MultiHead$_{MLP}^{(l)}(\cdot)$, respectively, in the $l$-th Transformer block ("Block-MLP"). Note that we omit the pre-layer normalization and dropout for simplicity.

In our experiments involving three shortcut-connected positions, models configured with "Pos-2" achieve the highest accuracy in both vision and language cases, while also ensuring substantial overlap duration. As a result, we favor selecting "Pos-2" for practical development. Moreover ,the "Pos-2" configuration is used to elucidate the overlapping strategy in Section 3.2. The specifics of the other two configurations can be inferred by analogy.

Additionally, our proposed ScMoE architecture can be adapted to support MoE models with varying MoE placement frequencies. As illustrated in Figure 5, the ScMoE architecture can be integrated into MoE models that incorporate an MoE module into every Transformer block. With more frequent MoE placement, the potential overlap duration for each MoE module is minimized, and the "Pos-1" configuration has already fully utilized the computation duration for the overlap. Conversely, less frequent MoE placement extends the potential overlap duration for each MoE module, which may lead to increased acceleration.

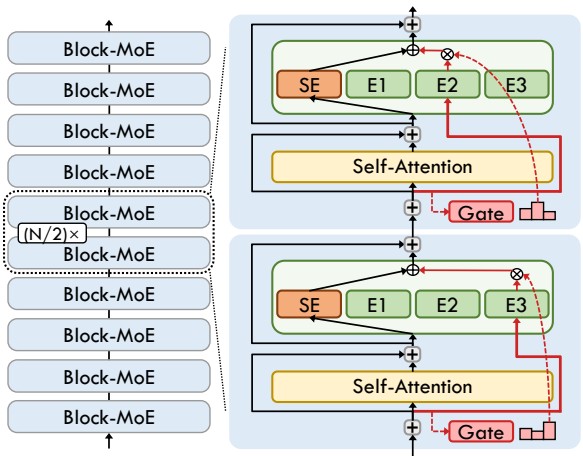

*Figure 5.* Illustration showcasing the application of the ScMoE (Pos-1) architecture to the MoE model, wherein the MoE module is integrated into each Transformer block.

## 3.2. Overlapping Strategy for Expert Parallelism

As mentioned in the previous section, the MoE operations in ScMoE architecture are completely decoupled from the backbone network, enabling parallel execution across two independent streams: one for the shared expert process and the other for the MoE process. To enhance efficiency, we implement asynchronous All-to-All communication operators to enable the overlapping of communication and computation within these streams, while computation operators are unable to execute concurrently due to the constraints on computing resources.

**Adaptive Operators Scheduling.** We observe that operator execution times are influenced by the specific model and hardware configuration, necessitating the implementation of adaptive scheduling for operators.

Following the execution order in the MoE stream, we can directly schedule the gate routing and encode operators at the earliest viable position while deferring the decode operator to the latest position, thereby maximizing the potential duration for overlapping. Then, this challenge is distilled into the selection of an optimal position for expert computation among four possible locations ①②③④ within the shared expert stream, as depicted in Figure 6.

Formally, we define the communication costs associated with "All-to-All Dispatch" and "All-to-All Combine" as $\mathcal{T}_{disp}$ and $\mathcal{T}_{comb}$, respectively. The variable $\mathcal{K}$ is designated to represent the specific location where expert computation is applied. Prior to the expert computation, the computational costs are denoted as $\mathcal{T}_{comp}^{pre} := \{COMP_1, ..., COMP_{\mathcal{K}-1}\}$, while the costs following the expert computation are represented as $\mathcal{T}_{comp}^{post} := \{COMP_{\mathcal{K}+1}, ..., COMP_4\}$. Consequently, the minimal aggregate time cost for each pair consisting of one Block-

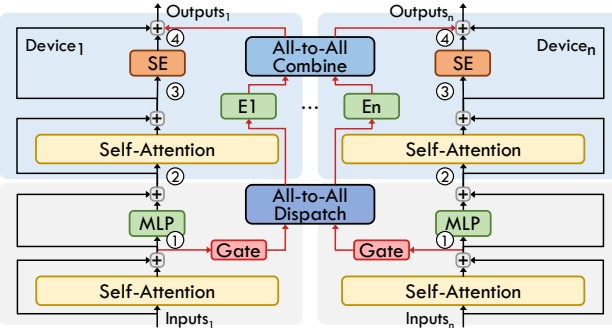

*Figure 6.* An overview of advanced expert parallelism using our proposed ScMoE architecture and overlapping strategy. The red line represents the decoupled MoE stream and the numbers ① through ④ denote the potential locations for expert computation.

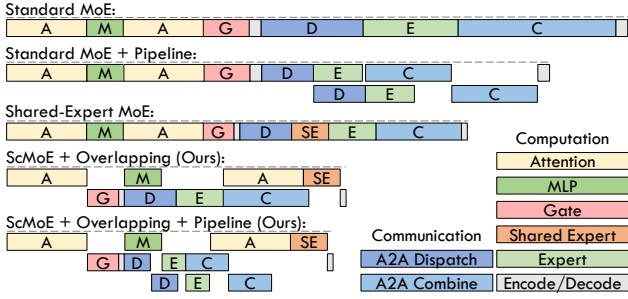

*Figure 7.* The timeline of different MoE architectures with corresponding parallel strategies, including pipeline and our proposed overlapping strategy. In each timeline, the length of each operator represents its time cost, while multiple rows indicate the utilization of parallel CUDA streams. The standard MoE utilizes top-2 gating, whereas the shared-expert MoE and ScMoE activate one shared expert alongside one gate-routed expert.

MLP and one Block-MoE is

$$\mathcal{T}_{overall}^{block} = \min_{\mathcal{K}}(|\mathcal{T}_{comp}^{pre} - \mathcal{T}_{disp}| + |\mathcal{T}_{comp}^{post} - \mathcal{T}_{comb}|)$$
$$= \min_{\mathcal{K}}(|\sum_{i=1}^{\mathcal{K}-1} COMP_i - \mathcal{T}_{disp}| + |\sum_{i=\mathcal{K}+1}^{4} COMP_i - \mathcal{T}_{comb}|),$$
(11)
$$\mathcal{T}_{overall}^{block} \geq |(\mathcal{T}_{comp}^{pre} + \mathcal{T}_{comp}^{post}) - (\mathcal{T}_{disp} + \mathcal{T}_{comb})|, \quad (12)$$
$$\mathcal{T}_{overall}^{block} \leq (\mathcal{T}_{comp}^{pre} + \mathcal{T}_{comp}^{post}) + (\mathcal{T}_{disp} + \mathcal{T}_{comb}). \quad (13)$$

To demonstrate the efficiency, we have illustrated the operational timelines of various MoE architectures alongside their respective parallel strategies in Figure 7, exemplified by the selection of location ② for expert computation. Each timeline's operator length corresponds to its execution time, and the presence of multiple rows signifies the utilization of parallel CUDA streams.

The widely-used pipeline parallel strategy equally segments input tokens into smaller fine-grained chunks, enabling concurrent computation and communication dispatched on dis-

tinct GPU streams (Hwang et al., 2023; Zhang et al., 2023). Contrary to standard MoE with pipelining (*2nd timeline*), our proposed ScMoE with the overlapping strategy (*4th timeline*) significantly reduces the total time. This reduction is attributed to the decrease in absolute communication time, similar to that in the shared-expert MoE (*3rd timeline*), and the overlap of communication with the computation duration ($\mathcal{T}Atten + \mathcal{T}SE + \mathcal{T}_{MLP}$), which extends beyond the overlap duration achieved through pipelining.

Our strategy possesses the capability to fully overlap communication if the communication can be accommodated within the overlapping window. This advantage is not shared by the pipeline strategy as it cannot overlap the initial and terminal data transmissions (Huang et al., 2019; Narayanan et al., 2019). In cases where communication durations exceed the available overlap duration, our strategy can be augmented with pipelining (*5th timeline*), thus utilizing the expert computation duration to further hide communication.

## 4. Experiments

### 4.1. Experimental Setup

To assess the effectiveness of our proposed overlapping strategy for enhancing expert parallelism, we conduct experiments on three hardware configurations: 8×A30-PCIe, 8×A800-NVLink, and 16×A800-NVLink (across 2 nodes). These configurations cover scenarios with both high and low communication-to-computation ratios. Furthermore, we evaluated our methods using both vision models (SwinV2-MoE) (Hwang et al., 2023) and language models (GPT2-MoE, GPT3-MoE, LLaMA2-MoE) (Radford et al., 2019; Brown et al., 2020; Touvron et al., 2023). Additional details on the experimental setup are provided in Appendix A.8.

### 4.2. Analysis of Model Quality and Efficiency

In this section, we assess the quality of the models with our proposed ScMoE architecture. Furthermore, we evaluate the efficiency of ScMoE models in distributed scenarios, which are accelerated through our proposed overlapping strategy for enhancing expert parallelism. To maintain the same computational volume as the standard top-2 MoE, both the experimental shared-expert MoE and our ScMoE utilize one shared expert and one gate-routed expert.

#### 4.2.1. VISION MODEL

Table 1 shows that ScMoE (Pos-2) and the standard top-2 MoE attain a comparable accuracy of 79.3%, while the shared-expert MoE delivers the highest accuracy, with a marginal increase of 0.2%. In 8×A30-PCIe where communication overhead accounts for 60% of the total MoE time, ScMoE (Pos-2) exhibits 30% speed improvement in training and 40% in inference compared to the standard top-2 MoE.

*Table 1.* Test top-1 accuracy and end-to-end speedup of train and inference (one iteration) for SwinV2-MoE-S (Hwang et al., 2023) models with various architectures pre-trained on ImageNet-1K for 90 epochs in the 8×A30-PCIe scenario, using standard MoE with top-2 gating as the baseline.

| Model | ImageNet-1K (Acc@1↑) | Train (Speedup↑) | Inference (Speedup↑) |
|---|---|---|---|
| Standard top-2 MoE | 79.33% | 1 | 1 |
| Standard top-1 MoE | 78.95% | 1.27× | 1.39× |
| Shared-Expert MoE | **79.53%** | 1.24× | 1.35× |
| Our ScMoE (Pos-1) | 79.14% | 1.36× | 1.54× |
| Our ScMoE (Pos-2) | **79.38%** | 1.43× | 1.66× |
| Our ScMoE (Pos-3) | 79.20% | **1.49×** | **1.82×** |

In scenarios characterized by a high communication-to-computation ratio, where communication is hard to be completely overlapped by computation, ScMoE architectures with extended overlap durations can achieve superior speedup. Specifically, ScMoE (Pos-3), which has the longest overlap duration of $\mathcal{T}_{Atten} + \mathcal{T}_{SE} + \mathcal{T}_{MLP}$, achieves the highest acceleration, with a 1.49× speedup in training and a 1.82× speedup in inference. Furthermore, the three different ScMoE architectures result in minimal variations in accuracy, ranging from 79.14% to 79.38%. Additionally, these methods, which utilize two activated experts, consistently outperform the standard top-1 MoE in terms of model quality, as the top-1 approach activates fewer parameters.

#### 4.2.2. LANGUAGE MODEL

To demonstrate the effectiveness of our proposed ScMoE architecture in models with two prevailing MoE placement frequency designs, we perform experiments using the GPT2-MoE-Medium and LLaMA2-MoE models, positioning the MoE module in every second Transformer block for GPT2-MoE and in every Transformer block for LLaMA2-MoE. Specifically, we utilize the ScMoE architecture with the configuration of "Pos-2" on the GPT2-MoE model, since this setup yields the lowest final validation loss in our experiments across various shortcut-connected positions, as discussed in Section 5.2. Furthermore, the ScMoE in LLaMA2-MoE adopts the "Pos-1" configuration, as this setup has already maximized the potential overlap duration in the scenario of MoE placement in every block.

As shown in Table 2, our ScMoE models achieve the highest average scores, with 38.69 in GPT2-MoE and 38.96 in LLaMA2-MoE. Furthermore, when integrated with our ScMoE, GPT2-MoE experienced an 11% improvement in training speed and an 15% improvement in inference speed compared to the standard top-2 MoE, in the 8×A800-NVLink scenario where communication accounts for 15% of the total MoE time. In LLaMA2-MoE, our ScMoE accelerated training by 1.14× and inference by 1.21×, demonstrating superior efficiency compared to other methods.

*Table 2.* Comparison of zero-shot evaluation and end-to-end speedup of training and inference (one iteration) for the pre-trained GPT2-MoE and LLaMA2-MoE models with various architectures in the 8×A800-NVLink scenario, using standard top-2 MoE as the baseline.

| Model | Method | Train | Inference | HellaSwag | PIQA | WinoGrande | BoolQ | ARC-E | OBQA | RACE | MathQA | AVG.(↑) |
|---|---|---|---|---|---|---|---|---|---|---|---|---|
| GPT2-MoE | Standard top-2 | 1 | 1 | 27.53 | 59.19 | 48.62 | 59.72 | 38.43 | 25.20 | 23.83 | 20.37 | 37.86 |
| | Shared-Expert | 1.04× | 1.06× | 27.23 | 59.09 | 51.22 | 60.00 | 38.85 | **26.60** | **25.07** | **20.57** | 38.58 |
| | Our ScMoE | **1.12×** | **1.17×** | **27.70** | **59.25** | **52.09** | **60.76** | **39.23** | 25.40 | 24.98 | 20.10 | **38.69** |
| LLaMA2-MoE | Standard top-2 | 1 | 1 | 28.40 | 60.07 | 50.83 | 58.26 | 38.72 | 24.60 | 25.17 | **21.07** | 38.39 |
| | Shared-Expert | 1.06× | 1.11× | 29.08 | 60.01 | 50.91 | **60.92** | 38.59 | 24.80 | 25.45 | 20.80 | 38.82 |
| | Our ScMoE | **1.14×** | **1.21×** | **29.09** | **60.55** | **51.38** | 57.25 | **38.89** | **26.40** | **27.08** | 21.07 | **38.96** |

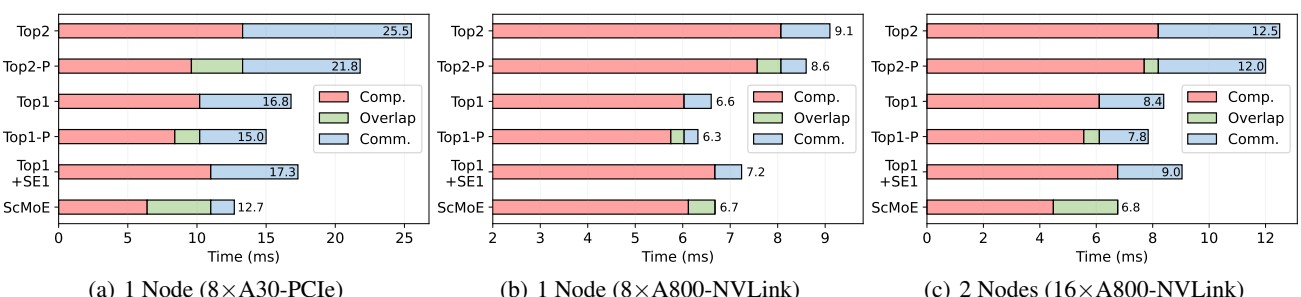

(a) 1 Node (8×A30-PCIe)          (b) 1 Node (8×A800-NVLink)          (c) 2 Nodes (16×A800-NVLink)

*Figure 8.* Overhead analysis for each pair of Block-MLP and Block-MoE within SwinV2-MoE-S model, deployed across three different distributed scenarios. "Topk" denotes the standard top-k MoE, while the one followed by the suffix "P" indicates using pipeline optimization as implemented by Tutel (Hwang et al., 2023). "Top1+SE1" refers to the shared-expert MoE.

### 4.2.3. ANALYSIS OF OVERHEAD AND ACCELERATION

In addition to exhibiting the end-to-end speedup of ScMoE in Tables 1 and 2, we delve into a detailed analysis of the overhead and the acceleration effect with our overlapping strategy, which can be generalized to other MoE models.

In the communication-intensive 8×A30-PCIe scenario (Figure 8(a)), our ScMoE overlaps 70% communication time, resulting in a 27% speed improvement over shared-expert MoE, a 42% improvement over the pipelined standard top-2 MoE, and a 15% improvement over the pipelined standard top-1 MoE. In the 8×A800-NVLink scenario (Figure 8(b)), which features almost minimal communication overhead, our approach maintains its acceleration by fully overlapping.

In multi-node scenario (Figure 8(c)), with 16×A800-NVLink across two nodes, communication incurs more significant overhead than in the single-node 8×A800-NVLink scenario due to the lower-bandwidth inter-node Ethernet (Li et al., 2020). Here, our ScMoE achieves complete overlap, resulting in a 24% speed improvement over the shared-expert MoE, a 43% improvement over the pipelined standard top-2.

In general, our ScMoE delivers a significant acceleration over the standard top-2 MoE, and even outperforms the top-1 MoE when communication exceeds approximately 20% of the total MoE time. Additionally, our ScMoE can fully overlap communication in scenarios where communication does not exceed an estimated 50% of the total MoE time.

## 5. Discussion

The empirical results presented in Section 4.2 have demonstrated that our proposed ScMoE architecture facilitates efficiency optimizations without compromising model quality. Subsequently, we delve into a more thorough examination of the proposed shortcut connection, uncovering potential underlying reasons for its algorithmic effectiveness, and identifying opportunities for further development.

### 5.1. Delve into the Proposed Shortcut Connection

#### 5.1.1. ANALYSIS OF GATING BEHAVIORS

Firstly, we investigate the use of the same MoE module to select the top-1 expert twice for processing each input token's current-layer and preceding-layer representations, respectively. As illustrated in Figure 9(a), we observe that the same gating network typically selects the same expert for the two representations of most tokens. As the training progresses, the token percentage of repeating selection initially escalates, peaking at 98%, and then diminishes, with a significant drop manifested in the last MoE sub-layer.

Next, we measure the L2 distance (similarity) between each token's preceding-layer and current-layer representations. Figure 9(b) illustrates that, as training advances, the L2 distance initially decreases with network depth, then increases, and ultimately reaches its maximum value in the final layer. Since the gating network is used to classify the representa-

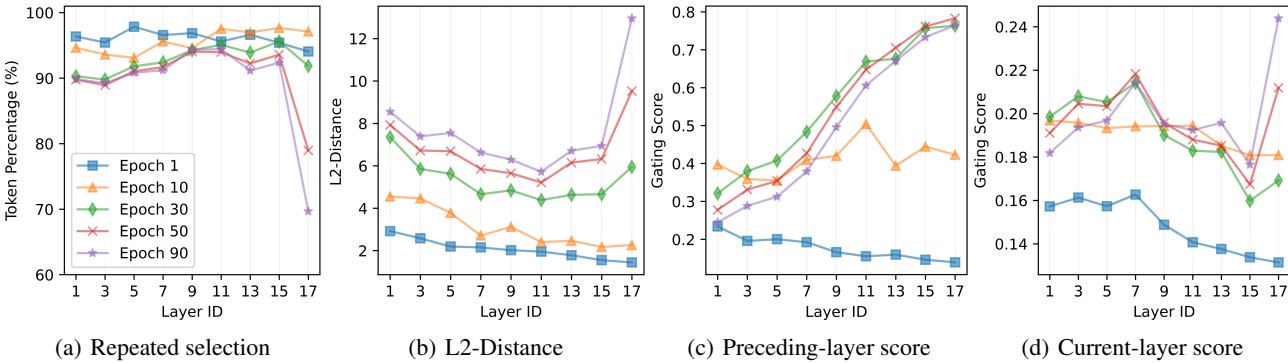

(a) Repeated selection       (b) L2-Distance       (c) Preceding-layer score       (d) Current-layer score

*Figure 9.* Results from the analysis of the proposed shortcut connection, during the 90-epoch training (including a 10-epoch warm-up) of the SwinV2-MoE-S model (Hwang et al., 2023). Employing the same MoE module to select the top-1 expert twice for processing each input token's two representations from the current and preceding layers, respectively, (a) illustrates the percentage of tokens that retain the same expert selection across the current layer and preceding layer, (b) shows the L2 distance between these two representations. Using the DGMoE, which imposes a constraint against repeatedly selecting the same expert, (c) presents the average gating score for the preceding-layer representations, (d) displays the average gating score for the current-layer representations.

tions, this similarity may lead to the repeated selection of the same experts, as evidenced by the correlation between the results in Figure 9(a) and 9(b).

Furthermore, we trained an experimental MoE model incorporating specialized MoE modules that select the top-1 expert twice for processing each input token's current-layer and preceding-layer representations. We observe that this experimental architecture achieves a model quality equivalent to the standard top-1 MoE, despite incurring the same computational cost as the top-2 MoE. Interestingly, this architecture can achieve model quality comparable to the standard top-2 MoE by imposing a constraint on the MoE module that ensures the selection of a different expert for the current layer than for the preceding layer. We refer to this enhanced experimental architecture as DoubleGating MoE (DGMoE), with further details provided in Appendix A.2. With this constraint, we observe gating score behaviors are similar to those of the standard top-2 MoE (Riquelme et al., 2021), as illustrated in Figure 9(c) and 9(d).

### 5.1.2. ANALYSIS OF SIMILARITY IN REPRESENTATIONS

Based on the observations mentioned above, we believe that the similarity between each token's preceding-layer and current-layer representations is crucial to understanding these outcomes and validating the effectiveness of our proposed ScMoE model. Assuming that the representations of the preceding layer and the current layer are identical, utilizing the same expert to process these two representations is equivalent to employing a single expert to process only the current-layer representations, thereby resulting in model quality comparable to that of the standard top-1 MoE. On the other hand, assigning distinct experts to the two representations of each token is equivalent to activating two experts to process the current-layer representations, thereby achieving model quality similar to that of the standard top-2.

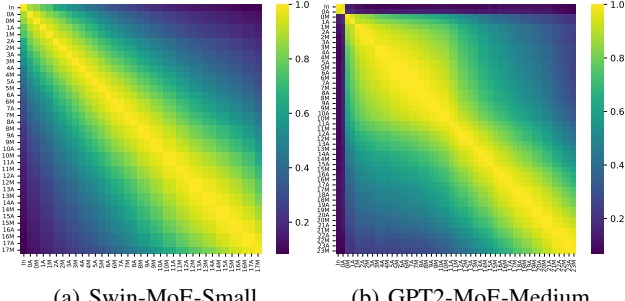

(a) Swin-MoE-Small       (b) GPT2-MoE-Medium

*Figure 10.* Analysis of cosine similarity in intermediate representations. The representations include the input to the first layer (denoted as 'In') and the outputs of the Attention (e.g., '1A') and MLP/MoE (e.g., '1M') sublayers within each Transformer block.

Moreover, we analyze the similarities in the intermediate representations of the Swin-MoE-Small and GPT2-MoE-Medium models, which use the standard top-2 MoE, as illustrated in Figure 10(a) and 10(b). It is evident that the representations from adjacent Transformer blocks exhibit a cosine similarity close to 1.0, highlighting their high degree of similarity. Consequently, our proposed ScMoE architecture assigns distinct experts to the two representations of each token (a shared expert for current-layer representations and routed experts for preceding-layer representations), thereby preserving behavior akin to the standard top-2 and shared-expert MoE architectures and ensuring comparable model quality. Similar observations in LLaMA2-MoE and OLMoE (Muennighoff et al., 2024), as demonstrated in Appendix A.9, further confirm the generalizability of our ScMoE to other models.

In addition, we provide a theoretical analysis of our proposed ScMoE architecture in Appendix A.1, elucidating the propagation of gradients to guarantee consistent training and preserve model quality.

## 5.2. Configuration of ScMoE Architecture

**Coefficient Gating Network.** In contrast to the gate-routed expert, the shared expert is fixed to process all representations without computing a gating score through the gating network. Therefore, some work (Qwen, 2024; Rajbhandari et al., 2022) employs a coefficient gating network $CG$ to generate the coefficient for combining the outputs of gate-routed and shared experts. Specifically, the coefficient gating network is a linear layer that uses the MoE module's input representation as its input to generate the coefficient.

We conduct experiments on ScMoE using three distinct methods for combining the outputs of gate-routed experts and shared experts: (1) Direct Add, (2) $CG$-1, and (3) $CG$-2. The Direct Add method, indicated by Equation 6, involves directly summing the outputs from both the shared expert and the gate-routed expert. For each input token $x \in \mathbb{R}^n$, the MoE outputs using $CG$-1 and $CG$-2, which generate coefficients for the combination, can be expressed as follows:

*CG-1:*
$$\text{coef} = Sigmoid(W_{\text{CG}} \cdot x), \quad W_{\text{CG}} \in \mathbb{R}^{1 \times n}, \quad (14)$$

$$MoE(x) = \text{coef} \cdot SE(x) + \sum_{i=1}^{N} G(x)_i E_i(x), \quad (15)$$

*CG-2:*
$$\text{coef} = softmax(W_{\text{CG}} \cdot x), \quad W_{\text{CG}} \in \mathbb{R}^{2 \times n}, \quad (16)$$

$$MoE(x) = \text{coef}[0] \cdot SE(x) + \text{coef}[1] \cdot \sum_{i=1}^{N} G(x)_i E_i(x). \quad (17)$$

As shown in Table 3, the configuration of $CG$-1 achieves the lowest final validation loss among the three combination methods, all of which are set to Pos-2. Moreover, ScMoE models with three configurations consistently outperform both the standard top-2 and the shared-expert MoE (configured with $CG$-2 according to (Rajbhandari et al., 2022)).

**Shortcut-connected Position.** While the Pos-1 configuration can maximize the potential overlap duration when MoE is placed in every block, using different configurations of shortcut-connected positions (Pos-1, Pos-2 or Pos-3) when placing MoE in every second block will result in variations in overlap duration and model quality. Therefore, we conduct experiments to identify its optimal configuration.

As shown in Table 3, ScMoE (Pos-2) achieves the lowest final validation loss among the configurations tested, all of which utilize the $CG$-1 setup. This outcome mirrors the findings from the vision experiments detailed in Table 1, where Pos-2 also delivers the highest accuracy.

As illustrated in Figure 10(b), the input and intermediate representations of the first layer differ significantly from those of subsequent layers, with the MLP/MoE altering the representations more substantially than Attention. Therefore, we explore modifying the first MoE module to utilize

*Table 3.* Comparison of the final validation loss of GPT-2 MoE pre-training across various MoE methods and configurations.

| MoE Method | Configuration | Final Validation loss ($\downarrow$) |
|---|---|---|
| Standard top-2 MoE | - | 3.270405 |
| Shared-Expert MoE | - | 3.240592 |
| Our ScMoE (Pos-2) | Direct Add | 3.236811 |
| | $CG$-1 | **3.224763** |
| | $CG$-2 | 3.232943 |
| Our ScMoE ($CG$-1) | Pos-1 | 3.237530 |
| | Pos-2 | **3.224763** |
| | Pos-3 | 3.241349 |
| | Pos-2 (L0 Pos-1) | 3.225626 |

Pos-1 while the remaining modules employ Pos-2, a configuration referred to as Pos-2 (L0 Pos-1). This setup results in a slightly higher loss compared to Pos-2 alone. Observations of varying shortcut-connected positions reveal that the superior performance of Pos-2 suggests the model quality of ScMoE is not necessarily improved by connecting more similar or dissimilar intermediate representations.

Consequently, we select the ScMoE configuration of $CG$-1 and Pos-2 for the experimental GPT2-MoE model, with its evaluations presented in Table 2.

### 5.3. Optimization for Memory-Limited Inference

Existing studies (Hwang et al., 2024; Yi et al., 2023) offload expert parameters to CPU memory in memory-limited inference scenarios where GPU cannot store the full MoE model. These studies utilize information in preceding layers to predict expert selection for the current MoE layer, enabling early expert migration from CPU to GPU and overlapping it with model computation. In contrast to existing speculative expert migration methods, we implement an expert offloading strategy with overlapping determinate migration, built upon our ScMoE that inherently advances expert selection to the preceding layer. The experimental results demonstrate that our expert offloading strategy reduces peak GPU memory usage by up to 60% and decreases expert migration costs by up to 75% through overlapping with computation. More details are shown in Appendix A.3.

## 6. Conclusion

The inherent dependency between communication and computation in conventional distributed MoE models hinders parallel optimization techniques to improve execution efficiency. To address this, we propose a shortcut-connected MoE (ScMoE) architecture, and develop a communication overlapping parallel strategy. Through empirical evaluation and theoretical analysis, our approaches demonstrate better execution efficiency while maintaining or exceeding the model quality of existing methods. In addition, we provide an insightful analysis and discussion of ScMoE.

## Acknowledgements

We thank the anonymous reviewers for their valuable comments. This work was supported in part by the Guangdong Provincial Project (No. 2023QN10X252), the Guangdong Basic and Applied Basic Research Foundation (No. 2023A1515110353), and GDIC. This research was conducted on the High-Performance Computing Platform of HKUST(GZ).

## Impact Statement

This paper presents work whose goal is to advance the field of Machine Learning. There are many potential societal consequences of our work, none which we feel must be specifically highlighted here.

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

# A. Appendix

## A.1. Theoretical Analysis

In this section, we delve deeper into the understanding of our proposed shortcut-connected MoE (ScMoE) architecture, presenting a theoretical foundation focused on the propagation of gradients to guarantee consistent training and preserve model quality. Our analysis is confined to the ScMoE (Pos-2) architecture as depicted in Figure 4(b); however, the same principles and derivations can be easily extended to other shortcut-connected MoE architectures. Building upon Equations 7 to 10, we can derive

$$
\begin{aligned}
\mathcal{H}_{l+1} = \mathcal{H}_l^{MH} + \Big( & \text{MLP}^{(l)}(\mathcal{H}_l^{MH}) \\
& + \text{MultiHead}^{(l+1)}(\mathcal{H}_l^{MH} + \text{MLP}^{(l)}(\mathcal{H}_l^{MH})) \\
& + \text{SE}^{(l+1)}(\mathcal{H}_l^{MH} + \text{MLP}^{(l)}(\mathcal{H}_l^{MH}) \\
& + \text{MultiHead}^{(l+1)}(\mathcal{H}_l^{MH} + \text{MLP}^{(l)}(\mathcal{H}_l^{MH}))) \\
& + \sum_{i=1}^{N} G(\mathcal{H}_l^{MH})_i E_i(\mathcal{H}_l^{MH}) \Big),
\end{aligned}
\tag{18}
$$

$$
\mathcal{H}_l^{MH} = \mathcal{H}_{l-1} + \text{MultiHead}^{(l)}(\mathcal{H}_{l-1}).
\tag{19}
$$

It is observable that Equations 18 and 19 share an identical structural expression. Consequently, we consider each pair of Block-MoE and Block-MLP layers as a single entity, and every sub-layer, denoted as $\mathcal{F}$, with its corresponding parameters $\mathcal{W}_l$, conforms to the equation

$$
x_{l+1} = x_l + \mathcal{F}_{\mathcal{W}_l}(x_l).
\tag{20}
$$

Here, $x_l$ represents the input, and $x_{l+1}$ represents the output of the $l$-th sub-layer. By applying this relationship recursively, the output of the uppermost $L$-th sub-layer, $x_L$, can be deduced as follows

$$
x_L = x_l + \sum_{i=l}^{L-1} \mathcal{F}_{\mathcal{W}_i}(x_i).
\tag{21}
$$

Let's consider the loss function as $\mathcal{E}$. Using the chain rule, we can calculate the derivative of the loss with respect to $x_l$, and we have

$$
\frac{\partial \mathcal{E}}{\partial x_l} = \frac{\partial \mathcal{E}}{\partial x_L} \frac{\partial x_L}{\partial x_l} = \frac{\partial \mathcal{E}}{\partial x_L} \Big( 1 + \frac{\partial}{\partial x_l} \sum_{i=1}^{L-1} \mathcal{F}_{\mathcal{W}_i}(x_i) \Big).
\tag{22}
$$

It's clear that the additive component of the error gradient $\frac{\partial \mathcal{E}}{\partial x_L}$ ensures direct information propagation back to any sub-layer $x_l$. Additionally, its advantage is that the number of product elements on the right side is independent of the network's depth. Therefore, as $L$ increases, it is less likely to encounter the gradient vanishing or exploding problem, ensuring stable training and sustained performance levels in our proposed MoE architectures.

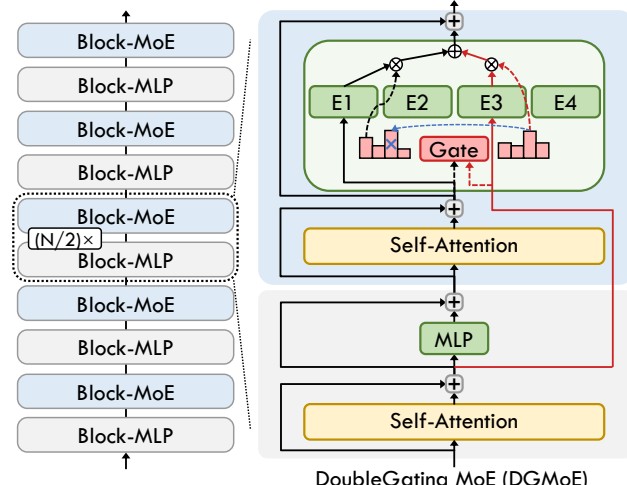

*Figure 11.* Illustration of the experimental DoubleGating MoE (DGMoE) architecture.

## A.2. Analysis of the DoubleGating MoE (DGMoE)

To delve deeper into our architecture with shortcut connection, we introduce the DoubleGating MoE (DGMoE) architecture, which employs dual top-1 gating mechanisms to independently process the representations from the preceding and current layers, as illustrated in Figure 11. Building upon Equations 7 to 10, and contrasting with ScMoE, DGMoE can be formulated as

$$
\begin{aligned}
\mathcal{H}_{l+1}^{\text{DGMoE}} = \mathcal{H}_{l+1}^{MH} + \sum_{i=1}^{N} \big( & G(\mathcal{H}_{l+1}^{MH})_i E_i(\mathcal{H}_{l+1}^{MH}) \\
& + G(\mathcal{H}_l^{MH})_i E_i(\mathcal{H}_l^{MH}) \big),
\end{aligned}
\tag{23}
$$

where $\mathcal{H}_{l+1}^{\text{DGMoE}}$ refers to the output from the MoE module.

However, as delineated in Equation 23, a potential issue arises when a token at the current layer selects the same top-1 expert as the preceding layer, inadvertently collapsing the intended top-2 gating mechanism into a de facto top-1 gating mechanism. To mitigate this, we introduce a constraint that ensures the activation of two distinct experts. In practice, this is achieved by first documenting the indices of experts triggered by the preceding-layer representations. Subsequently, if the preceding-layer representation coincidentally targets the same expert as the current layer, that is, if $\overline{TopK}(H(\mathcal{H}_l^{MH}), 1) = \overline{TopK}(H(\mathcal{H}_{l+1}^{MH}), 1)$, we activate the second-highest-ranking expert from the top-2 selection for the current layer, *i.e.*, $\overline{TopK}(H(\mathcal{H}_{l+1}^{MH}), 2)_2$.

As illustrated in Table 6 and Table 7, our DGMoE achieves comparable accuracy to the standard top-2 MoE across both vision and language tasks. Meanwhile, our ScMoE demonstrates performance more akin to the shared-expert MoE.

## A.3. Shortcut-connected MoE for Optimizing Memory-Limited Inference

While MoE effectively enhances LLMs in terms of model quality, it faces significant deployment challenges during on-device inference due to high memory demand. A common approach is to offload expert parameters to CPU memory (Shen et al., 2022) in scenarios where GPU memory is insufficient to store the entire MoE model. Moreover, decoder-only models use an autoregressive process for natural language generation (NLG) inference tasks, allowing for per-token processing of MoE. Specifically, only the two activated experts (top-2 gating) for each token need to be transferred from CPU to GPU memory for computation, thereby reducing peak GPU memory usage.

Since the migration of activated expert parameters from CPU to GPU, which occurs after expert selection, blocks expert computation until the transfer is complete, existing studies (Hwang et al., 2024; Yi et al., 2023; Du et al., 2024) have explored prefetching the experts. For instance, Pre-gated MoE (Hwang et al., 2024) uses information from preceding layers to predict expert selection, allowing for preloading of expert parameters into GPU memory, as shown in Figure 12 (a). This method enables overlapping the expert migration duration with the computation of preceding modules. Moreover, speculative expert migration methods adjust only the expert selection process, while expert computation continues along the same data flow of representations as in standard MoE.

However, speculative expert migrations can suffer from estimation inaccuracies, as they deviate from the original logic of pre-trained models, potentially reducing inference accuracy. In contrast, our proposed ScMoE architecture utilizes the gate-routed expert to compute the preceding-layer representations, inherently facilitating early expert migration well before the expert computation in the current layer. This allows us to implement an expert offloading strategy with overlapping determinate migration, maintaining the pre-trained logic.

Additionally, existing expert migration methods cannot be adapted to overlap communication in expert parallelism. This is because they do not decouple dependencies in the data flow of expert processing representations, and therefore cannot adjust the All-to-All communication of these representations

### A.3.1. EXPERT OFFLOADING STRATEGY

We implement an expert offloading strategy that keeps non-expert and shared expert modules in GPU memory while offloading other gate-routed experts to CPU memory. After the Attention module in the preceding layer generates intermediate representations, the gate determines expert se-

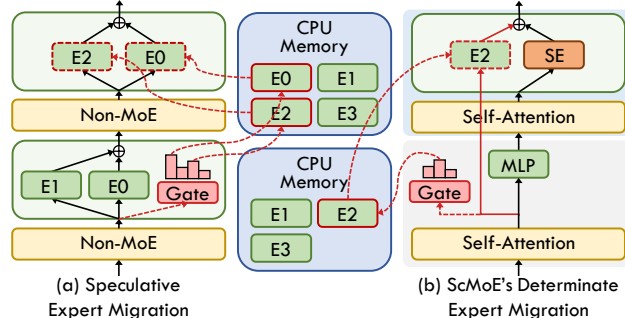

(a) Speculative Expert Migration  (b) ScMoE's Determinate Expert Migration

*Figure 12.* Illustrations of various expert migration methods to improve the efficiency of expert offloading: (a) speculative expert migration, exemplified by Pre-gated MoE (Hwang et al., 2024), and (b) our ScMoE's determinate expert migration. The red dashed line indicates expert selection and the transfer of expert parameters from CPU memory to GPU memory, while the black or red solid lines represent the data flow of representations processed by the Attention, MLP, and expert modules.

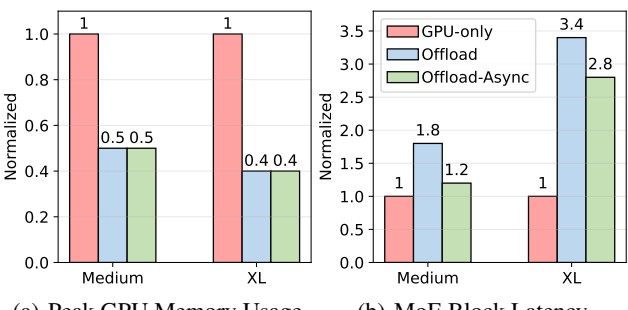

(a) Peak GPU Memory Usage  (b) MoE Block Latency

*Figure 13.* Peak GPU memory usage (a) and MoE block latency (b) for various memory-limited inference methods applied to the GPT2-MoE-Medium (8 experts per MoE module) and GPT3-MoE-XL models using ScMoE. "GPU-only" indicates that the entire model is stored in GPU memory. "Offload" refers to our strategy of offloading expert parameters to CPU with blocking expert migration. "Offload-Async" denotes the use of asynchronous expert migration to overlap its duration.

lection and issues asynchronous migration of the activated expert, as illustrated in Figure 12(b). This approach allows expert migration to overlap with the computation duration. Importantly, expert selection in our method adheres to the logic of the pre-trained ScMoE model, without speculation.

### A.3.2. EVALUATION

We evaluate our proposed expert offloading strategy on models with our ScMoE (Pos-2) architecture, using a platform with a single A30-PCIe GPU. As demonstrated in Figure 13(a), our expert offloading strategy reduces peak GPU memory usage by 50% for the GPT2-MoE-Medium model and by 60% for the GPT3-MoE-XL model when deployed in the inference scenario using a single A30-PCIe GPU. Furthermore, it is anticipated that models with more

*Table 4.* Comparison of validation perplexity and end-to-end speedup analysis of train and inference (one iteration) for our pre-trained GPT3-MoE-XL (Brown et al., 2020) models with various architectures in 8×A800-NVLink scenario, using standard MoE with top-2 gating as the baseline. "ScMoE-2" refers to the activation of one shared expert and two gate-routed experts.

| Model | Validation (Perplexity↓) | Train (Speedup↑) | Inference (Speedup↑) |
|---|---|---|---|
| Standard top-2 | 17.52 | 1 | 1 |
| Our ScMoE | 16.46 | **1.12×** | **1.18×** |
| Standard top-3 | 17.26 | 0.94× | 0.92× |
| Our ScMoE-2 | **16.27** | 1.05× | 1.08× |

gate-routed experts in each MoE module will experience a larger percentage reduction in GPU memory usage.

Since the offloaded expert parameters must be loaded into the GPU memory for expert computation, the blocking execution of this expert migration results in significant overhead. As shown in Figure 13(b), the blocking expert migration introduces an additional overhead of 80% in GPT2-MoE-Medium and 240% in GPT3-MoE-XL, substantially increasing the MoE block latency. To mitigate this issue, our strategy of asynchronously executing the determinate expert migration effectively reduces the additional costs by 75% in GPT2-MoE-Medium and 25% in GPT3-MoE-XL.

Furthermore, it is evident that expanding the model size from Medium to XL significantly raises the cost proportion related to expert migration. This is because the per-token decoding process during inference is memory-bound (Patel et al., 2024; Wu et al., 2024). The larger model size leads to a proportional increase in the duration of memory transfer, without a corresponding increase in computation time.

### A.4. Analysis of More Activated Experts

As increasing the number of activated experts within standard MoE is correlated with enhancements in model quality, we implement this augmentation in our ScMoE by increasing the count of gate-routed experts that process the preceding-layer representations, while maintaining the process of current-layer representations. To investigate the benefits of more activated experts, we implement the ScMoE-2, which employs top-2 experts for the preceding layer and one shared expert for the current layer.

Comparative analyses with the standard top-3 MoE, which has the same computational volumes as our ScMoE-2, reveal that our ScMoE architectures maintain superiority in both model quality and efficiency, as evidenced in Table 4. Furthermore, akin to the standard MoE, our ScMoE consistently improves with additional expert activation, shown by a decrease in validation perplexity from 16.46 with ScMoE to 16.27 with ScMoE-2.

*Table 5.* Comparison of top-1 accuracy on the ImageNet-1K test set for SwinV2-MoE-S models, using Direct Add and $CG$-1.

| Model | $CG$-1 | Direct Add |
|---|---|---|
| Shared-Expert MoE | **79.53%** | **79.02%** |
| Our ScMoE (Pos-1) | 79.14% | 78.78% |
| Our ScMoE (Pos-2) | **79.38%** | **78.98%** |
| Our ScMoE (Pos-3) | 79.20% | 78.29% |

*Table 6.* Comparison of top-1 accuracy on the ImageNet-1K test set for SwinV2-MoE-S and SwinV2-MoE-B models with various architectures: top-2/top-1 gating standard MoE, shared-expert MoE, our DGMoE, and ScMoE, each pre-trained for 90 epochs on the ImageNet-1K classification dataset.

| Model | SwinV2-MoE-S (Acc@1↑) | SwinV2-MoE-B (Acc@1↑) |
|---|---|---|
| Standard top-2 MoE | 79.33% | 80.48% |
| Standard top-1 MoE | 78.95% | 80.05% |
| Shared-Expert MoE | **79.53%** | **80.62%** |
| Our DGMoE (Pos-2) | 79.35% | 80.51% |
| Our ScMoE (Pos-2) | **79.38%** | **80.56%** |

Although activating more experts incurs higher time costs, the efficiency improvements of our overlapping strategy remain significant. For instance, our ScMoE-2 requires merely 95% and 93% of the time cost necessary for the standard top-2 MoE respectively in training and inference, despite processing increased computational loads.

### A.5. Coefficient Gating Network in Vision Task

As shown in Table 5, the incorporation of the coefficient gating network significantly enhances model performance in our experimental vision tasks. In the absence of the coefficient gating network, the quality of MoE architectures with shared experts declines from that of a standard top-2 MoE to that of a standard top-1 MoE, despite maintaining the same computational volume as the standard top-2 MoE.

### A.6. Evaluation Across Different Model Sizes

Table 6 and Table 7 illustrate that our experimental MoE architectures consistently achieve analogous model quality across different model sizes, as expounded in the detailed analysis within the main body of this paper.

### A.7. Share MoE Across Multiple Layers via Shortcut Connections

From a certain point of view, our shortcut-connected MoE architectures can be conceptualized as the sharing of one MoE module across multiple transformer layers. Parameter sharing across different layers has been validated as a method to enhance parameter efficiency and improve model

Table 7. Comparison of zero-shot perplexity on WikiText-103 for our pre-trained GPT2-MoE-Small and GPT2-MoE-Medium (8 experts per MoE module) models with various architectures.

| Model | GPT2-MoE-Small (Perplexity↓) | GPT2-MoE-Medium (Perplexity↓) |
|---|---|---|
| Standard top-2 MoE | 31.60 | 19.18 |
| Shared-Expert MoE | **29.15** | **17.94** |
| Our DGMoE (Pos-2) | 31.52 | 18.91 |
| Our ScMoE (Pos-2) | **29.10** | **17.62** |

quality, as evidenced in existing research (Lan et al., 2019; Dehghani et al., 2018; Xue et al., 2022; Huang et al., 2017).

The empirical analysis of our novel MoE architectures suggests that the MoE modules shared across multiple layers via shortcuts could offer a more parameter-efficient solution. We conduct experiments on a preliminary architecture DGMoE-Share which shares a single MoE for two pairs of transformer blocks. It reduces the parameter count from 157M to 124M, while maintaining the same volume of expert computation as the standard top-1 MoE. The DGMoE-Share achieves a 78.45% accuracy on the vision task, incurring a minimal accuracy decrement of 0.5% relative to the standard top-1 MoE. We anticipate the discovery of more efficient architectures through future explorations. Additionally, the optimization of training hyperparameters for the shortcut-connected MoE requires more investigation.

## A.8. Experimental Details

**Hardware Configurations.** To assess the effectiveness of our proposed overlapping strategy for enhancing expert parallelism, we conducted experiments on three hardware configurations: 8×A30-PCIe, 8×A800-NVLink and 16×A800-NVLink (across 2 nodes). These configurations cover scenarios with both high and low communication-to-computation ratios. Additionally, we evaluate our proposed expert offloading strategy on a configuration with a single A30-PCIe GPU.

**Experiments on Vision Model.** To evaluate the efficacy of our MoE architectures on vision tasks, we conduct experiments on SwinV2-MoE model, which is a state-of-the-art vision transformer model built upon the Tutel MoE framework (Hwang et al., 2023; Liu et al., 2021). Specifically, we pre-train the SwinV2-MoE models with various MoE architectures on ImageNet-1K image classification dataset, and subsequently evaluate their accuracy on the corresponding test set. It is noteworthy that the integration of the MoE module within SwinV2 is confined to stages 3 and 4, with our architectural enhancements being selectively applied to the MoE modules in stage 3—the deepest submodel. Given our hardware constraints, we configure each MoE module with 8 experts, assigning one expert per GPU device.

Table 9 summarizes the hyperparameters for training the Swin-MoE models including SwinV2-MoE-S and SwinV2-MoE-B. Specifically, the experiments related to overhead and acceleration analysis in a 2-node (16×A800-NVLink) scenario utilize 16 experts per MoE module, while other cases use 8 experts. To maintain the comparability of our experiments, we limit our modifications solely to the MoE architectures and keep the hyperparameters and random seeds consistent. In addition, the experimental results related to efficiency are the averages of multiple samples over different periods.

**Experiments on Language Model.** For natural language generation (NLG) tasks, we utilize the standard implementations of GPT-2 (Radford et al., 2019), GPT-3 (Brown et al., 2020) and LLaMA-2 (Touvron et al., 2023) from Fairseq (Ott et al., 2019), augmented with Tutel MoE to construct GPT2-MoE, GPT3-MoE and LLaMA2-MoE models. Specifically, we implement GPT2-MoE and GPT3-MoE by substituting the MLP with MoE in the second Transformer block of every consecutive pair, while implement LLaMA2-MoE by by substituting the MLP with MoE in every Transformer block. For models undergoing zero-shot evaluation on downstream tasks such as HellaSwag (Zellers et al., 2019), PIQA (Bisk et al., 2020), WinoGrande (Sakaguchi et al., 2021), BoolQ (Clark et al., 2019), ARC-Easy (Clark et al., 2018), OpenBookQA (Mihaylov et al., 2018), RACE (Lai et al., 2017), and MathQA (Amini et al., 2019), we pre-train the models using various architectures on a 1B token subset of the SlimPajama-627B dataset (Soboleva et al., 2023). For models evaluated on WikiText-103 (Merity et al., 2017), we conduct pre-training with different architectures on the OpenWebtext dataset (Gokaslan & Cohen, 2019). Table 8 summarizes the hyperparameters for training the GPT2-MoE-Small, GPT2-MoE-Medium, GPT3-MoE-XL and LLaMA-MoE models.

*Table 8.* Hyperparameters for GPT-MoE and LLaMA2-MoE models.

| Parameter | GPT2-MoE-Small | GPT2-MoE-Medium | GPT3-MoE-XL | LLaMA2-MoE |
|---|---|---|---|---|
| Num. layers | 12 | 24 | 24 | 24 |
| Embedding dim | 768 | 1024 | 2048 | 2048 |
| Num. attention heads | 12 | 16 | 32 | 16 |
| Num. KV heads | 12 | 16 | 32 | 4 |
| Num. experts per layer | 8 | 16 | 8 | 8 |
| MoE frequency | 1/2 | 1/2 | 1/2 | 1 |
| Num. parameters | 323M | 1.7B | 4.1B | 6.7B |
| Context/sequence length | 1K | 2K | 2K | 2K |
| Capacity factor | 2.00 | 2.00 | 2.00 | 2.00 |
| MoE loss coefficient | 0.01 | 0.01 | 0.01 | 0.01 |

*Table 9.* Hyperparameters for SwinV2-MoE models.

| Parameter | SwinV2-MoE-S | SwinV2-MoE-B |
|---|---|---|
| Image size | 192×192 | 192×192 |
| Window size | 12×12 | 12×12 |
| Embedding dim | 96 | 128 |
| Num. layers | [ 2, 2, 18, 2 ] | [ 2, 2, 18, 2 ] |
| Num. attention heads | [ 3, 6, 12, 24 ] | [ 4, 8, 16, 32 ] |
| Num. experts per layer | 8/16 | 8 |
| Batch size | 1024 | 1024 |
| Epochs | 90 | 90 |
| Warmup epochs | 10 | 10 |
| Base LR | 1.25$e$-4 | 1.25$e$-4 |
| Warmup LR | 1.25$e$-7 | 1.25$e$-7 |
| Min LR | 1.25$e$-6 | 1.25$e$-6 |
| Capacity factor | 1.25 | 1.25 |
| MoE loss coefficient | 0.01 | 0.01 |

## A.9. Additional Examples of Intermediate Representations Similarities

*Figure 14.* Intermediate representation similarities in LLaMA2-MoE.

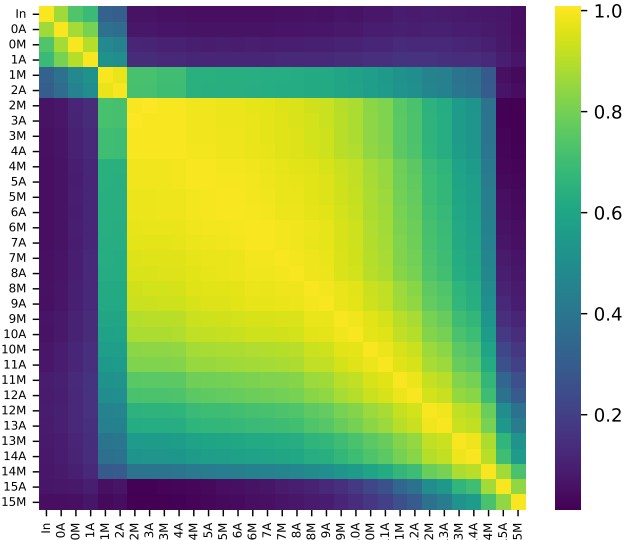

*Figure 15.* Intermediate representation similarities in OLMoE (Muennighoff et al., 2024).

