# OpenReview forum: "Shortcut-connected Expert Parallelism for Accelerating Mixture of Experts"
_ICML.cc/2025/Conference — ICML 2025 poster_

### Official Review · Reviewer_qsHP · 2025-02-20

**Overall Recommendation:** 3

**Summary:**

The execution of mixture-of-experts model contains two all-to-all communication steps on the critical path of computation. The authors of this paper propose to use the activations before the attention layer of the current block as the input for the experts in the current block, which breaks the sequential dependency between the attention and expert layers, and opens the possibility to overlap the computation and communication. Experiments on the accuracy and inference time show the effectiveness of the new design.

**Claims And Evidence:**

Yes

**Essential References Not Discussed:**

No

**Experimental Designs Or Analyses:**

Yes, the experimental design is good in general. But it's better to include more modern designs like the ones used in deepseek v2 (more smaller experts).

**Methods And Evaluation Criteria:**

Yes

**Other Comments Or Suggestions:**

No

**Other Strengths And Weaknesses:**

**Strengths**
1. The paper addresses an important problem: reducing communication overhead in the serving of MoE models.
2. By modifying the architecture to break the sequential dependency between attention and MoE layers, the authors enable overlapping computation and communication, effectively hiding communication overhead.
3. Experiments demonstrate that the proposed computation-communication overlap strategy in the new architecture is effective.
4. The paper is well-written and easy to follow.

**Weaknesses**
My primary concern is that the skip connection design lacks novelty. Additionally, the model quality requires further justification or experiments to substantiate that it's actually effective. Existing models such as GPT-J and GPT-NeoX already parallelize attention and feedforward networks, while architectures like Snowflake Arctic [1] employ a structure where the input to the self-attention layer is directly fed to the MoE layer - closely resembling this work’s proposal. However, current SOTA models (e.g., DeepSeek V3/R1, Llama 3.2) do not adopt this parallel structure, raising questions about whether the proposed architecture can achieve competitive accuracy/quality at scale. The experiments in this paper inadequately validate model quality, as the metrics in Table 2 fall significantly below those of publicly available models, including smaller ones.

In general, altering model architecture for performance optimization but do not validate the final accuracy on large scale carry risks, as such changes may compromise final accuracy. The accuracy number of Snowflake Arctic [1] also shows that this parallel architecture might harm the model quality (when it compares with other models with similar size of parameters). While the paper is well-argued on the motivation and the system-side optimizations appear sound, I remain cautious about the final model quality of the new architecture.

[1] https://www.snowflake.com/en/blog/arctic-open-efficient-foundation-language-models-snowflake/

**Questions For Authors:**

No

**Relation To Broader Scientific Literature:**

A new design to MoE architecture.

**Theoretical Claims:**

No

---

> ### Author Rebuttal · Authors · 2025-03-31
>
> > 1. It's better to include more modern designs like the ones used in deepseek v2 (more smaller experts).
>
> Given our constraints on available hardware, we cannot conduct experiments on large-scale MoE models like DeepSeek-V2 (236B). Nonetheless, we conduct experiments on the OLMoE model (7B), a recent MoE model that features more modern designs, incorporating the fine-grained expert architecture (more smaller experts) similar to DeepSeek-V2. We train both the original OLMoE and the OLMoE+ScMoE from scratch, using 40,000 samples from the TuluV3 dataset. As demonstrated in the following table, our proposed ScMoE architecture not only accelerates the training by 1.13x and inference by 1.19x, but also achieves a marginally higher average accuracy (43.77% vs 43.25%).
>
> | |Train Speedup|Inference Speedup|ARC-C|BoolQ|OBQA|RTE|WG|PIQA|AVG|
> |-|-|-|-|-|-|-|-|-|-|
> |OLMoE|1|1|24.91%|58.30%|30.00%|44.77%|49.64%|51.90%|43.25%|
> |OLMoE+ScMoE |1.13x|1.19x|25.51%|58.10%|30.80%|45.49%|49.25%|53.48%|43.77%|
>
>
>
> > 2. My primary concern is that the skip connection design lacks novelty. Additionally, the model quality requires further justification or experiments to substantiate that it's actually effective. Existing models such as GPT-J and GPT-NeoX already parallelize attention and feedforward networks, while architectures like Snowflake Arctic [1] employ a structure where the input to the self-attention layer is directly fed to the MoE layer - closely resembling this work’s proposal.
>
> (1) GPT-J and GPT-NeoX utilize parallel Attention and Feed-Forward Layers to address the All-Reduce communication overhead inherent in Tensor Parallelism, which is notably different from the All-to-All communication overhead encountered in Expert Parallelism. Distinct from existing methods, our proposed ScMoE architecture is specifically designed to minimize the All-to-All communication overhead in Expert Parallelism.
>
> (2) Regarding Snowflake Arctic, our work was conducted independently and concurrently with it, and it stands out in several distinct ways. Snowflake Arctic is primarily an industrial product and does not include any published papers or detailed analyses of their proposed Dense-MoE Hybrid Transformer architecture. In contrast, we thoroughly examine the effectiveness of Shortcut-connected MoE architectures and analyze the phenomena and reasons behind them, offering deeper insights for the research community. Furthermore, our proposed Shortcut-connected MoE demonstrates superior generalization across different MoE model designs (adapt to various MoE placement frequencies). It bears similarity to their Dense-MoE Hybrid Transformer architecture, specifically to our proposed ScMoE (Pos-1), which is more like a subset of our study.
>
> > 3. However, current SOTA models (e.g., DeepSeek V3/R1, Llama 3.2) do not adopt this parallel structure, raising questions about whether the proposed architecture can achieve competitive accuracy/quality at scale. The experiments in this paper inadequately validate model quality, as the metrics in Table 2 fall significantly below those of publicly available models, including smaller ones. In general, altering model architecture for performance optimization but do not validate the final accuracy on large scale carry risks, as such changes may compromise final accuracy. The accuracy number of Snowflake Arctic [1] also shows that this parallel architecture might harm the model quality (when it compares with other models with similar size of parameters). While the paper is well-argued on the motivation and the system-side optimizations appear sound, I remain cautious about the final model quality of the new architecture.
>
> (1) We propose the ScMoE architecture as a viable option for the backbone of MoE models. In line with the initial experiments conducted on previously proposed MoE architectures, such as standard MoE and shared-expert MoE, we train the ScMoE model from scratch to ensure a fair comparison. When trained under identical conditions, ScMoE demonstrates superior efficiency while achieving comparable accuracy, compared to both the standard MoE and shared-expert MoE methods.
>
> (2) Due to our limited devices, we are unable to train with pre-training datasets comprising 1 trillion tokens, as is common with existing industrial-level models. Consequently, our experimental models exhibit relatively lower accuracy, as we only train with datasets that are 1/1000th the size. Nevertheless, we offer theoretical analysis upon our empirical results to support the effectiveness of our methods, as discussed in Section 5 and Appendix A.1.
>
> (3) According to the provided document on Snowflake Arctic, this parallel architecture offers significant value for practical deployments, rather than solely impacting model quality negatively. The document states that "Arctic is more capable than other open source models trained with a similar compute budget."

---

> > ### Comment · Reviewer_qsHP · 2025-04-04
> >
> > Thanks the authors for addressing my concerns. I agree that the parallel industrial work without publication should not be a blocker for this paper. Changing my score from 2 to 3.

---

> > > ### Author Response · Authors · 2025-04-07
> > >
> > > Thank you for your thoughtful review and for improving the score. If you have any further concerns or questions, we would be more than happy to address them.

---

### Official Review · Reviewer_hvmz · 2025-03-10

**Overall Recommendation:** 3

**Summary:**

The paper proposes an new method for expert parallelism, which is a paradigm for distributed training and inference of large scale MoE models by dividing experts across multiple devices. The authors address the bottleneck of all-to-all communication between experts and present a new strategy that can overlap communication with computation. Given a model with every other layer being an MoE and each MoE having a shared expert, the shortcut-connected approach overlaps MLP and shared expert computation with the all-to-all communications of routed experts. This results in tangible speedups for MoE training and inference for GPT2-size models.

**Claims And Evidence:**

The main premise of reducing expert parallelism overhead is well motivated. The proposed method is clear and the paper shows convincing evidence that the proposed overlapping strategy is beneficial for improving throughput. However, the MoE architecture is fundamentally changed in this method. This leads to slightly decreased performance on pretraining benchmarks, and it is not clear whether this error scales for larger models.

**Essential References Not Discussed:**

N/A

**Experimental Designs Or Analyses:**

The experimental design makes sense. The authors apply their new MoE architecture for LM pretraining on multiple models, up to 7B parameters. I think the analysis can be stronger if all of the the results are consolidated across different model sizes. For example, what is the % speedup, peak GPU memory reduction, change in validation loss, change in pretraining benchmarks as a result of model size? Currently the evaluation is sparse as the benchmarks in Table 2 are reported for two models and the latency + peak memory analysis in Figure 13 uses two different models. Because the paper is fundamentally modifying the MoE architecture it would be more convincing to see all of the evaluations for each model.

**Methods And Evaluation Criteria:**

The evaluation section is very strong and uses sensible benchmarks. The most common pretraining benchmarks are reported along with validation loss. The paper also discusses the speedup of their method, which makes sense as the goal of their method was to reduce MoE distributed communication overhead.

**Other Comments Or Suggestions:**

N/A

**Other Strengths And Weaknesses:**

The method seems limited to architectures that use shared experts and alternating MoE layers with typical MLPs. While this is a commonly used setup, I think the results would be more convincing if the approach could be extended to architectures that don't follow these exact constraints.

The approach also fundamentally modifies the MoE architecture by introducing these shortcut connections. I assume this means that ScMoE cannot be applied to speed up MoE inference for off-the-shelf pretrained models, which limits its wider adoption.

**Questions For Authors:**

1. How do the validation loss results in Figure 10 compare with a regular MoE as the baseline?
2. How does the overall speedup and benchmark performance scale with model size? Do you expect that the ScMoE approach becomes more viable for larger scale pretraining?
3. Can this shortcut connection approach be applied to already pretrained models? What does the performance look like in such cases? Does your approach necessarily require pretraining a model from scratch to use the shortcut connections
4. How do you think your results could generalize to other MoE models that either do not use shared experts, or have an MoE for every transformer layer?

I think the results of this paper are very interesting with strong impact on the field of MoE research in general. However, they currently seem limited to a specific architecture setup, and I am not convinced if this is viable for MoE training at larger scale. The responses to these questions would increase my confidence in recommending this paper for acceptance.

**Relation To Broader Scientific Literature:**

The authors address the bottlenecks expert parallelism, which is becoming an increasingly important paradigm for training and deploying large scale MoEs. The results in this paper have the potential to speed up MoE training in a broad range of settings, which makes this work widely applicable in the field of MoE research.

**Theoretical Claims:**

I did not check for correctness of any proofs

---

> ### Author Rebuttal · Authors · 2025-03-31
>
> > 1. I think the analysis can be stronger if all of the the results are consolidated across different model sizes.
>
> We have summarized and supplemented evaluations of GPT models across various sizes to better demonstrate the impact of model size as a standalone variable. The following  table presents the speedup, zero-shot perplexity, peak GPU memory reduction, and MoE block latency reduction of ScMoE in comparison to the standard top-2 MoE. The results demonstrate that our proposed ScMoE consistently achieves improvements across various model sizes.
>
> | |Model Size|Train Speedup|Inference Speedup|Perplexity (ScMoE / Standard MoE) |Peak GPU Memory Usage Reduction|MoE Block Latency Reduction|
> |-|-|-|-|-|-|-|
> |GPT2-MoE-Small| 323M|1.15x|1.22x|29.10/31.60|-53%|-41%|
> |GPT2-MoE-Medium|1.7B|1.12x|1.17x|17.62/19.18|-50%|-33%|
> |GPT2-MoE-XL|4.1B|1.12x|1.18x|16.46/17.52|-60%|-18%|
>
> > 2. How do the validation loss results in Figure 10 compare with a regular MoE as the baseline?
>
> We observed that the loss curve of ScMoE (3.2629 at the final point) demonstrates faster convergence and a lower final loss compared to the standard top-2 MoE (3.3157 at the final point) and shared-expert MoE (3.2974 at last).
>
> > 3. How does the overall speedup and benchmark performance scale with model size? Do you expect that the ScMoE approach becomes more viable for larger scale pretraining?
>
> (1) To illustrate the effectiveness of speedup in larger-scale scenarios, we conducted additional simulations, the results of which are detailed in **the table "Simulation of larger-scale model training" in our response to Reviewer GE4M**. The findings suggest that the relative ratio of communication to computation duration, influenced by various training and model configurations, is crucial for impacting speedup. Optimal speedup is achieved when these durations are balanced. Despite these variations, ScMoE consistently achieves acceleration.
>
> (2) Due to our limited devices, we face challenges in conducting experiments on large-scale MoE models. Nevertheless, we offer theoretical analysis to support the performance of our methods, as discussed in Section 5 and Appendix A.1.
> Additionally, based on the findings presented in Question 1, we anticipate that the disparity in accuracy between ScMoE and existing MoE architectures will diminish as both model size and the volume of training data increase.
>
> > 4. Can this shortcut connection approach be applied to already pretrained models? What does the performance look like in such cases? Does your approach necessarily require pretraining a model from scratch to use the shortcut connections.
>
> (1) We propose the ScMoE architecture as a viable option for the backbone of MoE models. In line with the initial experiments conducted on previously proposed MoE architectures, such as standard MoE and shared-expert MoE, we train the ScMoE model from scratch to ensure a fair comparison.
>
> (2) The ScMoE models can be constructed using pre-trained shared-expert MoE models by modifying the inputs of the MoE module. Our experiments with this approach showed a decrease in accuracy, indicating a need for further refinements in the fine-tuning process to preserve accuracy.
>
> (3) Additionally, it is possible to construct and train a ScMoE model based on a pretrained dense model, employing techniques similar to sparse upcycling, which have been applied to standard MoE models.
>
> > 5. The method seems limited to architectures that use shared experts and alternating MoE layers with typical MLPs. How do you think your results could generalize to other MoE models that either do not use shared experts, or have an MoE for every transformer layer?
>
> (1) We would like to clarify that we have already conducted experiments on the current mainstream MoE structure of placing MoE module in every Transformer block, as presented by the LLaMA2-MoE experiments in Section 4.2.2, Table 2 and Table 7. Specifically, our LLaMA2-MoE experiments utilize ScMoE (Pos-1) and incorporates the MoE module in every Transformer block, which overlaps the communication and computation within the each Transformer block.
>
> (2) Our work focuses on integrating shortcut connections into the MoE architecture to enhance efficiency, which does not rely on the shared-expert MoE. As discussed in Appendix A.2, our DGMoE, which incorporates shortcut connection without shared expert, also offers improvements over the top-2 MoE.
>
> Furthermore, ScMoE functions alongside both the standard MoE and shared-expert MoE architectures. When designing models, practitioners have the option to choose from these three architectures.
>
> (3) We conduct experiments on the OLMoE model, which originally employs standard MoE without shared experts at each layer. As shown in **the table "Experimental results on integrating ScMoE architecture into the OLMoE model" in response to Reviewer GE4M**, our proposed ScMoE architecture not only achieves acceleration, but also achieves a higher average accuracy.

---

### Official Review · Reviewer_rMhT · 2025-03-11

**Overall Recommendation:** 3

**Summary:**

This paper presents ScMoE to enhance the computational efficiency of Mixture-of-Experts (MoE) models. By incorporating a shortcut connection that integrates information from the preceding layer with the current layer's computations, ScMoE introduces a concurrent processing mechanism which allows for overlapping communication with computation through the parallel execution of two independent streams: one processing the current layer's input and another propagating residual from the previous layer. As a result, ScMoE significantly accelerates both the training and inference phases by minimizing communication latency while maintaining or even enhancing model quality.

**Claims And Evidence:**

Communication and computation can overlap fully depending greatly on hardware and system conditions.

**Essential References Not Discussed:**

The references are comprehensive and well-organized.

**Experimental Designs Or Analyses:**

The experiments are limited to only small models, which may limit the generalizability of the results to other models or tasks.

**Methods And Evaluation Criteria:**

The claim that model quality is always maintained or improved is only shown on small models. Sometimes the accuracy improvements are very small, and the results might not work for larger models (like OLMoE) or different tasks.

**Other Comments Or Suggestions:**

The authors have only conducted experiments on small models for image classification and language tasks, so additional experiments on larger and more powerful models may be necessary.

**Other Strengths And Weaknesses:**

The proposed method is simple, which is a clear advantage. However, it relies heavily on the assumption that features in neighboring layers are similar, which is often true. Yet, the supplementary materials show that this similarity is not clear in the OLMoE model. In addition, the authors mention that ScMoE-Pos2 does not perform well, which makes me worry about how well the method can work in different cases.

**Questions For Authors:**

Does the proposed method fail if the features change dramatically between neighboring layers?

**Relation To Broader Scientific Literature:**

The authors demonstrate that neighboring Transformer layers share significant feature similarity, a pattern consistent with prior research.

**Theoretical Claims:**

The theoretical claims presented in the paper are solid.

---

> ### Author Rebuttal · Authors · 2025-03-31
>
> > 1. The claim that model quality is always maintained or improved is only shown on small models. Sometimes the accuracy improvements are very small, and the results might not work for larger models (like OLMoE) or different tasks. The experiments are limited to only small models, which may limit the generalizability of the results to other models or tasks. The authors have only conducted experiments on small models for image classification and language tasks, so additional experiments on larger and more powerful models may be necessary.
>
> (1) Given our constraints on available hardware, we encounter difficulties in conducting empirical experiments on large-scale, industrial-level MoE models. Nevertheless, we perform additional experiments on the OLMoE model [1] and conduct some different evaluation tasks. The detailed experimental configurations are introduced in our response to Reviewer GE4M. As shown in the following table, our proposed ScMoE architecture not only accelerates the training by 1.13x and inference by 1.19x, but also achieves a marginally higher average accuracy (43.77% vs 43.25%).
>
> |             | Train Speedup | Inference Speedup | ARC-C  | BoolQ  | OBQA   | RTE    | WG     | PIQA   | AVG    |
> | ----------- | ------------- | ----------------- | ------ | ------ | ------ | ------ | ------ | ------ | ------ |
> | OLMoE       | 1             | 1                 | 24.91% | 58.30% | 30.00% | 44.77% | 49.64% | 51.90% | 43.25% |
> | OLMoE+ScMoE | 1.13x         | 1.19x             | 25.51% | 58.10% | 30.80% | 45.49% | 49.25% | 53.48% | 43.77% |
>
> (2) Beyond the focus on accuracy, we also simulations to validate the effectiveness of speedup in larger-scale scenarios, as shown in **the table "Simulation of larger-scale model training" in our response to Reviewer GE4M**. Despite differences in communication and computation durations across configurations, our proposed ScMoE consistently proves to be effective.
>
> > 2. It relies heavily on the assumption that features in neighboring layers are similar, which is often true. Yet, the supplementary materials show that this similarity is not clear in the OLMoE model. In addition, the authors mention that ScMoE-Pos2 does not perform well, which makes me worry about how well the method can work in different cases. Does the proposed method fail if the features change dramatically between neighboring layers?
>
> (1) Recent studies [2,3,4,5] have commonly observed the high similarity between features in neighboring layers as a characteristic of Transformer-based LLMs, which can be utilized to improve the efficiency of LLM operations.
>
> (2) Our supplementary materials on the OLMoE model further corroborate that the observed feature similarity is within an expected range and aligns with prevalent findings in the field. As illustrated in Figure 15, most data points near the diagonal line are yellow, indicating nearly 100% similarity. Only the output features of 1A and 1M exhibit comparatively lower similarity, yet maintain a significant 50%.
>
> (3) We would like to clarify that the statement "ScMoE-Pos2 does not perform well" was made in a comparative context. In fact, the loss differences among ScMoE-Pos1 (3.2615), ScMoE-Pos2 (3.2818), and ScMoE-Pos2-L0Pos1 (3.2629) are relatively minor. Moreover, the loss values for the standard top-2 MoE (3.3157) and shared-expert MoE (3.2974) are higher than those of the three ScMoE configurations. This indicates that despite variations in feature similarity across certain layers, ScMoE remains effective. Additionally, we will include a detailed comparison of these loss curves in the appendix for further clarity.
>
> (4) Our strategy for selecting shortcut-connected positions, detailed in Section 5.1.3, is designed to achieve optimal accuracy. Additionally, based on these observations of similarity, we can adapt this strategy for subsequent MoE model design by opting not to use ScMoE in the first two Transformer layers. Similarly, DeepSeek-MoE [6] chooses not to apply the MoE module in the first Transformer layer.
>
> **Reference**
>
> [1] Muennighoff, Niklas, et al. "Olmoe: Open mixture-of-experts language models." *arXiv preprint arXiv:2409.02060* (2024).
>
> [2] Sun, Qi, et al. "Transformer layers as painters." arXiv preprint arXiv:2407.09298 (2024).
>
> [3] He, Shwai, et al. "What matters in transformers? not all attention is needed." arXiv preprint arXiv:2406.15786 (2024).
>
> [4] Men, Xin, et al. "Shortgpt: Layers in large language models are more redundant than you expect." *arXiv preprint arXiv:2403.03853* (2024).
>
> [5] Gromov, Andrey, et al. "The unreasonable ineffectiveness of the deeper layers." *arXiv preprint arXiv:2403.17887* (2024).
>
> [6] Dai, Damai, et al. "Deepseekmoe: Towards ultimate expert specialization in mixture-of-experts language models." *arXiv preprint arXiv:2401.06066* (2024).

---

### Official Review · Reviewer_f6vs · 2025-03-12

**Overall Recommendation:** 3

**Summary:**

This paper proposes Shortcut-connected MoE (ScMoE) to reduce the All-to-All communication bottleneck in expert parallelism of MoE model training. Traditional MoE models suffer from high All-to-All communication costs due to dependencies between computation and communication. ScMoE solves this by using a shortcut connection that allows the previous layer’s representations to be processed in parallel with the current layer. It replaces top-2 gating with a shared expert for the current layer and a top-1 expert for the previous layer, enabling full overlap between computation and communication. The authors also propose an adaptive overlapping scheduling strategy, achieving 1.49x faster training and 1.82x faster inference while keeping model accuracy comparable or slightly better than standard top-2 MoE. Experiments demonstrate that ScMoE maintains strong performance while improving efficiency. The paper also provides theoretical analysis proving stable gradient propagation. Overall, ScMoE co-designs algorithm and infrastructure, and is a novel and effective method for accelerating MoE training.

---

### update after rebuttal

After reading the rebuttal, my major concerns have been addressed.

**Claims And Evidence:**

Most of the claims are well-supported by experiments and analysis.

1. Decoupling and overlapping communication and computation via shortcut connection: The shortcut connection proposed by this paper does allow the overlapping between computation and communication, as demonstrated in Figure 5 and Figure 6. Experiments also confirm that ScMoE overlaps up to 100% of communication time, reducing the impact of All-to-All bottlenecks.
2. Maintaining Model Quality: The authors claim that the shortcut connection maintains the model quality. This is supported by the experiments, where ScMoE achieves 79.3% Top-1 accuracy on ImageNet, similar to top-2 MoE, and performs well in NLP tasks.
3. Generalization Capability: The experiments shows that ScMoE improves performance on different hardwares (A30-PCIe with high communication ratio, and A800-NVLink with low communication ratio), on different workloads (vision tasks, language tasks).

---

However, I still have the following concerns:
1. The author states that the standard MoE model structure consists of Block-MLP and Block-MoE, as shown in Figure 2, which aligns with the design in DeepSpeed-MoE. However, I am afraid such a structure is out-dated, and in most mainstream MoE models (e.g., DeepSeekMoE, DeepSeek-V3), MoE layers are predominantly arranged sequentially rather than interspersed with MLP layers. I am curious whether ScMoE is compatible with this mainstream MoE architecture.

**Essential References Not Discussed:**

Here are several related works:
- Switch Transformers: Scaling to Trillion Parameter Models with Simple and Efficient Sparsity [JMLR’22], which is the first to use top-1 gating for efficiency.
- DeepSeekMoE: Towards Ultimate Expert Specialization in Mixture-of-Experts Language Models [Arxiv’24], which is one of the mainstream MoE model structures.

**Experimental Designs Or Analyses:**

The experiments are well-controlled:
1. Diverse hardware setups: 8×A30 PCIe (high communication overhead) and 8×A800 and 16xA800 NVLink (low communication overhead).
2. Multiple Workloads: SwinV2-MoE-S (vision), GPT-2/LLaMA2-MoE (NLP).
3. Ablation studies: Different shortcut placements (Pos-1/2/3) to find the best trade-off between speed and accuracy.

---

However, I suppose the following evaluation will strengthen the analysis:
1. Evaluation on Mainstream MoE Structure: The workloads used in ScMoE is based on DeepSpeed-MoE, where MoE layers are interspersed with MLP layers. However, I think the current mainstream MoE structure (where MoE layers are predominantly arranged sequentially) should also be evaluated, e.g., DeepSeekMoE.
2. Stronger Baselines: It would be valuable to compare ScMoE with other advanced All-to-All optimization techniques, such as hierarchical All-to-All and pipelining. Additionally, DeepSeek recently open-sourced DeepEP, a high-performance expert parallelism library. While I understand that ScMoE is either concurrent with or an earlier work relative to DeepEP, including an analysis of their differences and potential complementarities would be beneficial.
3. Larger Models: Since ScMoE modifies the model structure, it may impact model convergence and overall quality. While the authors provide empirical results on models up to 7B and offer theoretical analysis, additional experiments on larger models (e.g., 13B, 30B) would strengthen the evaluation.
4. Larger Multi-node Cluster Analysis: I am concerned of the performance on larger multi-mode clusters. I understand authors may not be able to obtain larger scale multi-mode cluster (e.g., 64 A800 GPUs). However, it would be beneficial if authors provide analysis or simulated evaluation of the theoretical performance of ScMoE on larger clusters.

**Methods And Evaluation Criteria:**

The proposed ScMoE architecture and adaptive scheduling strategy effectively target the MoE communication bottleneck. The evaluation criteria makes sence, covering:
1. Benchmarks: Vision tasks (ImageNet) and NLP tasks (GPT-2/LLaMA2 MoE, with model size up to 7B).
2. Metrics: Model quality and accuracy, training efficiency, inference efficiency.
3. Baselines: Standard top-2 MoE (DeepSpeed MoE), Shared-expert MoE, and Standard top-1 MoE for comparison.
4. Hardwards: A30-PCIe with high communication ratio, and A800-NVLink with low communication ratio. For A800, 8 GPUs within 1 node and 16 GPUs across 2 nodes are evaluated.

**Other Comments Or Suggestions:**

See the questions for authors.

**Other Strengths And Weaknesses:**

Strengths:
1. Significant speedup without accuracy loss.
2. Well-structured experiments, with multiple baselines and ablation studies.
3. Good clarity and organization, making complex ideas easy to understand.

Weaknesses:
1. Incremental improvement: ScMoE mainly extends shared-expert MoE with shortcut connections.
2. Model structure may be out-dated: DeepSpeed-MoE, where MoE layers are interspersed with MLP layers, is not the mainstream MoE structure currently.
3. Lack of evaluation on larger-scale models and larger-scale multi-node cluster.

**Questions For Authors:**

Experiment Related:
1. Are GPT-2/LLaMA2 MoE models trained from scratch or fine-tuned from a dense model?
2. The experiments test single-node (8 GPUs) and two-node (16 GPUs) training. How does ScMoE scale to hundreds or thousands of GPUs? More analysis of ScMoE on multi-mode clusters would be helpful.
3. Are the experiments conducted on FP32 precision of BF16 precision?

---

Overlap Related:
1. Figure 6 provides a visual timeline comparison of different MoE architectures. Would it be possible to quantify and report exact percentages of communication hidden by computation in Table 1 or 2?
2. If we increase the number of experts per MoE layer, how does the overlap ratio change? Does a larger number of experts negatively impact ScMoE’s effectiveness?
3. What happens when communication is much slower than computation? Can ScMoE still provide significant speedup?

---

Model Quality Related:
1. Do you observe any convergence differences between ScMoE and top-2 MoE during training? If so, does ScMoE require different hyperparameters (learning rate, warm-up schedule, etc.)?
2. Have you tested transfer learning performance? For example, if a vision model is pre-trained with ScMoE and fine-tuned on a different task, does the shortcut mechanism still provide benefits?
3. Is ScMoE scalable to larger MoE models (e.g., 13B, 30B, 70B, and even DeepSeek-V3 671B)? I am concerned about the convergence and model quality.

**Relation To Broader Scientific Literature:**

ScMoE builds upon existing MoE parallelism research, including:
1. Expert Parallelism (Lepikhin et al., 2021), which introduces All-to-All communication for MoE.
2. Shared Expert MoE (Rajbhandari et al., 2022), which uses a fixed expert for every layer, reducing All-to-All communication.
3. Pre-gated MoE, which is optimized MoE inference by preselecting experts.

ScMoE combines shared experts with shortcut connections to further reduce communication overhead, making it a novel extension of these works.

**Theoretical Claims:**

The appendix provides a theoretical analysis of gradient propagation, showing that ScMoE does not affect model convergence. The authors prove that:
1. The shortcut connection preserves gradient flow between layers.
2. The shared expert and expert gating maintain stable optimization dynamics.

---

> ### Author Rebuttal · Authors · 2025-03-31
>
> > 1. Concerns regarding the model structure may be outdated, particularly in relation to the compatibility of ScMoE with the mainstream MoE architecture, where MoE layers are arranged sequentially rather than interspersed with MLP layers.
>
> We would like to clarify that we have already conducted experiments on the current mainstream MoE structure of placing MoE module in every Transformer block, as presented by the LLaMA2-MoE experiments in Section 4.2.2, Table 2 and Table 7. Specifically, our LLaMA2-MoE experiments utilize ScMoE (Pos-1) and incorporates the MoE module in every Transformer block, which overlaps the communication and computation within the each Transformer block.
>
> Additionally, we conduct experiments on OLMoE model, an advanced MoE model that integrates an MoE module at each layer. As shown in **the table "Experimental results on integrating ScMoE architecture into the OLMoE Model" in our response to Reviewer GE4M**, ScMoE architecture not only achieves acceleration, but also achieves a marginally higher average accuracy.
>
> > 2. Concerns regarding stronger baselines.
>
> We select a recently proposed All-to-All optimization technique, "MoE Shared Expert Overlap," which overlaps All-to-All communication with the computation of shared experts, as a stronger baseline for comparison.
>
> As shown in the following table, ScMoE achieves acceleration compared to "MoE Shared Expert Overlap," facilitated by breaking dependencies for improved overlap.
>
> |SwinV2-MoE|Train Speedup|Inference Speedup|
> |-|-|-|
> |Standard Top-2|0.76x|0.68x|
> |Shared-Expert|0.94x|0.92x|
> |MoE Shared Expert Overlap|1|1|
> |ScMoE|1.14x|1.25x|
>
> > 3. Concerns regarding larger multi-node cluster analysis. Does a larger number of experts negatively impact ScMoE’s effectiveness?
>
> To demonstrate scalability in large-scale clusters, we conducted simulations across different number of GPUs, from 16 to 128, as shown in **the table "Simulation of larger-scale model training" in our response to Reviewer GE4M**. Despite differences in communication and computation durations across configurations, ScMoE consistently demonstrates effectiveness. Notably, in entries No.5 to No.8, an increased number of experts does not negatively affect the effectiveness of ScMoE.
>
> > 4. Incremental improvement: ScMoE mainly extends shared-expert MoE with shortcut connections.
>
> Our work focuses on integrating shortcut connections into the MoE architecture to enhance efficiency, which does not rely on the shared-expert MoE. As discussed in Appendix A.2, our DGMoE, which incorporates shortcut connection without shared expert, also offers improvements over the top-2 MoE. ScMoE is presented as the most efficient architecture to facilitate our proposed shortcut connections.
>
> Furthermore, ScMoE functions alongside both the standard MoE and shared-expert MoE architectures. When designing models, practitioners have the option to choose from these three architectures.
>
> > 5. Are GPT-2/LLaMA2 MoE models trained from scratch or fine-tuned from a dense model?
>
> The models are trained from scratch. We introduce the ScMoE architecture as a viable option for pre-training a MoE model from scratch, enhancing the efficiency of its pre-training, fine-tuning, and inference.
>
> > 6. Are the experiments conducted on FP32 precision of BF16 precision?
>
> BF16 precision.
>
> > 7. Would it be possible to quantify and report exact percentages of communication hidden by computation in Table 1 or 2?
>
> The exact percentages for Table 1 are directly reflected in Figure 7 (1), which hides 70% communication.
>
> > 8. What happens when communication is much slower than computation? Can ScMoE still provide significant speedup?
>
> As shown by the simulation results of No. 1 in Question 3, ScMoE achieves a significant speedup of 1.31x, even when the communication time is three times longer than the computation duration.
>
> > 9. Do you observe any convergence differences between ScMoE and top-2 MoE during training? If so, does ScMoE require different hyperparameters (learning rate, warm-up schedule, etc.)?
>
> We observed that the loss curve of ScMoE (3.2629 at the final point) demonstrates faster convergence and a lower final loss compared to the standard top-2 MoE (3.3157 at the final point). To ensure a fair comparison in our experiments, ScMoE demonstrates comparable accuracy under identical hyperparameters.
>
> > 10. Have you tested transfer learning performance?
>
> We conduct additional fine-tuning experiments on the trained OLMoE models discussed in Question 1, using 5,000 samples from Alpaca dataset. The results shows that ScMoE still achieves a higher average accuracy (44.88% vs 44.07%).
>
> > 11. Is ScMoE scalable to larger MoE models (e.g., 13B, 30B, 70B, and even DeepSeek-V3 671B)?
>
> Due to our limited devices, we face challenges in conducting experiments on large-scale MoE models. Nevertheless, we offer theoretical analysis to support the effectiveness of our methods, as discussed in Section 5 and Appendix A.1.

---

### Official Review · Reviewer_GE4M · 2025-03-21

**Overall Recommendation:** 4

**Summary:**

MoE model is an effective way to scale up model parameters while preserving the inference latency. When training MoE models, due to the extremely large parameters, expert parallelism is widely used to distribute the computational workload. However, this also introduces expensive all-to-all communication cost. This paper proposes a new MoE architecture to overlap communication and computation, which helps to greatly save the training time. Small-scale experiments show that the new architecture can speedup the training while maintaining the same model performance as the original MoE architecture.

**Claims And Evidence:**

Yes

**Essential References Not Discussed:**

NA

**Experimental Designs Or Analyses:**

Yes

**Methods And Evaluation Criteria:**

I think the accuracy on benchmark datasets are a bit low to provide meaningful comparisons. It'd be better to train bigger models or longer.

**Other Comments Or Suggestions:**

NA

**Other Strengths And Weaknesses:**

Strengths:

- The idea of overlapping communication and computation in MoE training is novel and interesting. This could be a useful technique for practitioners.

Weakness:
- It would be great if the idea can be validated at larger model scale and longer training runs. Currently the accuracy on benchmarks are a bit low.

**Questions For Authors:**

NA

**Relation To Broader Scientific Literature:**

The authors propose a new MoE architecture which may be insightful to readers.

**Theoretical Claims:**

NA

---

> ### Author Rebuttal · Authors · 2025-03-31
>
> > 1. I think the accuracy on benchmark datasets are a bit low to provide meaningful comparisons. It'd be better to train bigger models or longer. It would be great if the idea can be validated at larger model scale and longer training runs. Currently the accuracy on benchmarks are a bit low.
>
> Given our constraints on available hardware, we encounter difficulties in conducting empirical experiments on large-scale, industrial-level MoE models. Nevertheless, we perform additional experiments on the OLMoE model [1], an advanced MoE model that activates 1B of its total 7B parameters. We initiate training from scratch for both the original OLMoE and the OLMoE+ScMoE models using 40,000 samples from the TuluV3 dataset [2]. As illustrated in the following table, our proposed ScMoE architecture not only accelerates the training by 1.13x and inference by 1.19x, but also achieves a marginally higher average accuracy (43.77% vs 43.25%).
>
> **Experimental results on integrating ScMoE architecture into OLMoE model [1].**
> |             | Train Speedup | Inference Speedup | ARC-C  | BoolQ  | OBQA   | RTE    | WG     | PIQA   | AVG    |
> | ----------- | ------------- | ----------------- | ------ | ------ | ------ | ------ | ------ | ------ | ------ |
> | OLMoE       | 1             | 1                 | 24.91% | 58.30% | 30.00% | 44.77% | 49.64% | 51.90% | 43.25% |
> | OLMoE+ScMoE | 1.13x         | 1.19x             | 25.51% | 58.10% | 30.80% | 45.49% | 49.25% | 53.48% | 43.77% |
>
> Moreover, we offer theoretical analysis upon our empirical findings to support the effectiveness of our methods. Specifically, we observe a correlation between the efficacy of ScMoE and the high similarity of intermediate representations. Given that such high similarity is commonly found in Transformer-based LLMs [3,4,5,6], we believe our ScMoE approach can be effective in larger-scale LLMs. Furthermore, our theoretical analysis of gradient propagation also demonstrates its ability to achieve consistent training while preserving model quality.
>
>
>
> Additionally, we conduct simulations to validate the effectiveness of speedup in larger-scale scenarios (larger model and larger number of GPUs). Our simulations employ the ASTRA-SIM simulator [7], which models an A800 cluster in alignment with the real-world HPC cluster utilized for evaluations in Section 4. Specifically, the FP16 computational capability is configured to 312 TFLOPS, the intra-node NVLink bandwidth is set at 400 GB/s, and the inter-node network employs a 200Gb/s (25GB/s Bandwidth) InfiniBand connection.
>
> We conduct simulations of the forward pass for LLaMA2-MoE, maintaining similar attention configurations to experiments in Table 2. Since the load balance of MoE significantly influences its efficiency and is determined by various inputs, we simulate a fully balanced distribution of input tokens to eliminate any random interference in this aspect.
>
> In the simulation, we vary the configurations for the number of GPUs, the number of active and total experts, and expert intermediate size. As illustrated in the following table, while different configurations lead to varying communication and computation durations, our overlapping strategies consistently prove to be effective.
>
> **Simulation of larger-scale model training.**
> |No.|GPU Num|Act./Total Experts|Expert Intermediate Size|Communication Time (us)|Computation Time (us)|Overlap Speedup|
> |-|-|-|-|-|-|-|
> |1| 128| top-16 / 128| 1024| 7056| 2169| 1.31x |
> |2| 64| top-8 / 64| 2048| 6345| 2968| 1.47x|
> |3| 32| top-4 / 32| 4096| 5994 | 4578| 1.76x|
> |4| 16| top-2 / 16| 8192 | 5832| 7805| 1.75x|
> |5| 128| top-16 / 128| 8192| 7056| 7819| 1.90x|
> |6| 64| top-8 / 64|8192| 6345| 7811| 1.81x |
> |7| 32| top-4 / 32|8192| 5994| 7807 | 1.77x |
> |8| 16| top-2 / 16| 8192 | 5832| 7805| 1.75x|
> |9| 16| top-8 / 128| 8192| 23229| 20725| 1.89x|
>
>
>
> **Reference**
>
> [1] Muennighoff, Niklas, et al. "Olmoe: Open mixture-of-experts language models." *arXiv preprint arXiv:2409.02060* (2024).
>
> [2] Lambert, Nathan, et al. "Tulu 3: Pushing frontiers in open language model post-training." arXiv preprint arXiv:2411.15124 (2024).
>
> [3] Sun, Qi, et al. "Transformer layers as painters." arXiv preprint arXiv:2407.09298 (2024).
>
> [4] He, Shwai, et al. "What matters in transformers? not all attention is needed." arXiv preprint arXiv:2406.15786 (2024).
>
> [5] Men, Xin, et al. "Shortgpt: Layers in large language models are more redundant than you expect." *arXiv preprint arXiv:2403.03853* (2024).
>
> [6] Gromov, Andrey, et al. "The unreasonable ineffectiveness of the deeper layers." *arXiv preprint arXiv:2403.17887* (2024).
>
> [7] Rashidi, Saeed, et al. "Astra-sim: Enabling sw/hw co-design exploration for distributed dl training platforms." ISPASS. IEEE (2020).

---

### Decision · Program_Chairs · 2025-05-01

**Decision:**

Accept (poster)

**Comment:**

None of the reviewers showed strong opinions either way, but all the reviews were at least borderline positive.

The method is intuitive, and the latency improvements make sense to me. The issue is in the empirical validation that the modification the authors are suggesting doesn't negatively impact the model. In Table 1, all of the proposed methods perform worse than the shared expert baseline. In Table 2, we see an Avg improvement across LM benchmarks of 0.01, which is well within the standard error, and the models are clearly undertrained (just 1B tokens of training data) because they are scoring near the noise floor for the benchmarks. This doesn't preclude acceptance because I have to imagine that the boost in training speed should translate into sufficiently more tokens that things are fine anyways.